# A novel monocyte differentiation pattern in pristane-induced lupus with diffuse alveolar hemorrhage

**Shuhong Han\*, Haoyang Zhuang, Rawad Daniel Arja, Westley H Reeves\***

Division of Rheumatology, Allergy, & Clinical Immunology, University of Florida, Gainesville, United States

**Abstract** Pristane causes chronic peritoneal inflammation resulting in lupus, which in C57BL/6 mice is complicated by lung microvascular injury and diffuse alveolar hemorrhage (DAH). Mineral oil (MO) also causes inflammation, but not lupus or DAH. Since monocyte depletion prevents DAH, we examined the role of monocytes in the disease. Impaired bone marrow (BM) monocyte egress in *Ccr2*−/− mice abolished DAH, confirming the importance of monocyte recruitment to the lung. Circulating Ly6C$^{hi}$ monocytes from pristane-treated mice exhibited increased annexin-V staining in comparison with MO-treated controls without evidence of apoptosis, suggesting that pristane alters the distribution of phosphatidylserine in the plasma membrane before or shortly after monocyte egress from the BM. Plasma membrane asymmetry also was impaired in Nr4a1-regulated Ly6C$^{lo/−}$ 'patrolling' monocytes, which are derived from Ly6C$^{hi}$ precursors. Patrolling Ly6C$^{lo/−}$ monocytes normally promote endothelial repair, but their phenotype was altered in pristane-treated mice. In contrast to MO-treated controls, Nr4a1-regulated Ly6C$^{lo/−}$ monocytes from pristane-treated mice were CD138$^+$, expressed more TremL4, a protein that amplifies TLR7 signaling, and exuberantly produced TNFα in response to TLR7 stimulation. TremL4 expression on these novel CD138$^+$ monocytes was regulated by Nr4a1. Thus, monocyte CD138, high TremL4 expression, and annexin-V staining may define an activated/inflammatory subtype of patrolling monocytes associated with DAH susceptibility. By altering monocyte development, pristane exposure may generate activated Ly6C$^{hi}$ and Ly6C$^{lo/−}$ monocytes, contributing to lung microvascular endothelial injury and DAH susceptibility.

**\*For correspondence:**
Shuhong.Han@medicine.ufl.edu (SH);
westley.reeves@medicine.ufl.edu (WHR)

**Competing interest:** The authors declare that no competing interests exist.

## Editor's evaluation

This is an interesting manuscript describing two subsets of Ly6Clo non-classical monocytes in pristane-treated mice, one CD138- and Nr4a1-independent and the other CD138+ and Nr4a1-dependent. The latter is believed to be produced during pristane-induced microvascular lung injury in an ineffectual effort to maintain vascular integrity in the face of ongoing endothelial damage. This is a model of lung injury with relevance to autoimmunity. Should this CD138+ Nr4a1-dependent population be a hallmark of vascular injury as the authors suggest this work could be relevant for many fields.

## Introduction

Monocytes are derived from bone marrow (BM) macrophage (Mφ) and dendritic cell progenitor cells (MDPs), which develop into common monocyte progenitor cells (cMoPs) and then Ccr2$^+$Ccr4$^{lo}$Ly6C$^+$ monocytes. Ccl2, a ligand for Ccr2, is essential for the egress of Ly6C$^+$ monocytes from the BM in response to inflammation (*Serbina and Pamer, 2006*; *Hanna et al., 2011*), after which these cells can develop into Ly6C$^{hi}$Ccr2$^{hi}$Cx3cr1$^{lo}$CD43$^−$ (Ly6C$^{hi}$) 'classical' monocytes or Ly6C$^{lo/−}$Ccr2$^{lo}$Cx3cr1$^{hi}$CD43$^+$ (Ly6C$^−$)

'non-classical' monocytes (*Yona et al., 2013*). CD62L (*Sell*), a marker expressed by newly emigrated BM-derived Ly6C$^+$ monocytes, is lost with maturation (*Ingersoll et al., 2010*; *Jakubzick et al., 2017*).

Ly6C$^-$ monocytes are derived from Ly6C$^{hi}$ precursors (*Yona et al., 2013*; *Sunderkötter et al., 2004*; *Varol et al., 2007*; *Sugimoto et al., 2015*). In unmanipulated C57BL/6 (B6) mice there are two populations of circulating monocytes: Ly6C$^{hi}$ (expressing *Ccr2*, *Lyz2*, *Sell*, *Irf4*, and other genes) and Ly6C$^-$ (expressing *Cx3cr1*, *Itgax*, *Bcl2*, *Pparg*, *Cd36*, *Cebpb*, *Nr4a1*, and other genes). A third population with intermediate expression of Ly6C (referred to here as Ly6C$^{lo}$) does not have a specific gene expression signature, but expresses genes characteristic of both Ly6C$^{hi}$ and Ly6C$^-$ monocytes (*Mildner et al., 2017*).

Differentiation of Ly6C$^-$ monocytes is driven by CCAAT-enhancer binding protein β (C/EBPβ) via activation of the transcription factor nuclear receptor subfamily 4 group A member 1 (Nr4a1) (*Hanna et al., 2011*; *Mildner et al., 2017*). Along with Nr4a1, Ly6C$^-$ monocyte development requires Notch signaling (*Hanna et al., 2011*; *Gamrekelashvili et al., 2016*). In contrast to the role of Ly6C$^{hi}$ monocytes as mediators of inflammation, Nr4a1-dependent Ly6C$^-$ monocytes patrol the vascular endothelium and promote the TLR7-dependent removal of damaged cells (*Auffray et al., 2007*; *Carlin et al., 2013*). Although they typically remain within blood vessels, they also can migrate across the endothelium into inflamed tissues (*Auffray et al., 2007*; *Schyns et al., 2019*). Ly6C$^-$ monocytes may represent a form of intravascular macrophage (Mφ) (*Ginhoux and Jung, 2014*), as suggested by the expression of genes typical of tissue-resident Mφ such as *Apoe* and *Cd36* (*Mildner et al., 2017*). Although only a single population of Ly6C$^-$ monocytes is seen in the blood of unmanipulated B6 mice, less is known about Ly6C$^-$ monocytes in inflammatory settings.

Pristane and mineral oil (MO) both cause sterile peritoneal inflammation and an influx of Ly6C$^{hi}$ monocytes, which become Ly6C$^{hi}$ inflammatory peritoneal Mφ (*Lee et al., 2009*). In pristane—but not MO—treated mice, the chronic inflammatory response evolves into lupus after several months (*Reeves et al., 2009*). In pristane-treated B6 mice, but not BALB/c or other strains, the onset of lupus is complicated by lung microvascular injury and diffuse alveolar hemorrhage (DAH) (*Zhuang et al., 2017*; H Zhuang, *In Revision*).

In contrast to the striking predominance of Ly6C$^{hi}$ Mφ in the peritoneum of pristane-treated mice, Ly6C$^{lo}$ Mφ predominate in MO-treated mice (*Han et al., 2017*). A subset of the Ly6C$^-$ peritoneal Mφ expresses CD138 (syndecan-1) (*Han et al., 2017*). Ly6C$^{lo}$CD138$^+$ Mφ from MO-treated mice have an anti-inflammatory phenotype and are highly phagocytic for dead cells. The significance of CD138 expression by Mφ is unclear and it is not known whether CD138 is expressed by monocytes. We examined the distribution of CD138 expression in myeloid cells from the blood and inflamed peritoneum of pristane- and MO-treated mice.

Peritoneal Ly6C$^{lo}$CD138$^+$ and Ly6C$^{lo}$CD138$^-$ Mφ both were Nr4a1 independent and Ccr2 dependent, suggesting they were derived from classical (Ly6C$^{hi}$) monocytes recruited to the peritoneum. Unexpectedly, in pristane-treated mice circulating Ly6C$^{-/lo}$ monocytes also consisted of CD138$^+$ and CD138$^-$ populations. However, only the Ly6C$^{-/lo}$CD138$^+$ subset was Nr4a1 dependent. Compared with the Ly6C$^{-/lo}$CD138$^-$ and Ly6C$^{hi}$ subsets, Ly6C$^{-/lo}$CD138$^+$ monocytes expressed higher levels of Triggering Receptor Expressed on Myeloid Cells Like 4 (Treml4), an innate immune receptor that amplifies TLR7 signaling and binds late apoptotic and necrotic cells (*Hemmi et al., 2009*; *Ramirez-Ortiz et al., 2015*; *Nedeva et al., 2020*; *Gonzalez-Cotto et al., 2020*). Treml4 expression was regulated by Nr4a1, consistent with a role in the Ly6C$^-$ monocyte-mediated, TLR7-dependent, removal of damaged endothelial cells. Circulating Nr4a1-dependent Ly6C$^-$CD138$^+$ monocytes increased substantially after pristane, but not MO, treatment, and were more abundant in B6 (DAH susceptible) vs. BALB/c (DAH resistant) mice. In contrast to pristane-treated mice, Nr4a1-dependent Ly6C$^{lo/-}$ monocytes in MO-treated mice were CD138$^-$. Thus, Nr4a1-dependent Ly6C$^{-/lo}$ 'patrolling' monocytes from pristane- vs. MO-treated mice were distinguishable by the presence or absence, respectively, of CD138 surface staining. Monocytes (both Ly6C$^{hi}$ and Ly6C$^{-/lo}$) from pristane- vs. MO-treated also were distinguished by the intensity of annexin-V staining, suggesting that pristane treatment causes a breakdown of plasma membrane asymmetry, which could have functional implications for the pathogenesis of pristane-induced lupus and alveolar hemorrhage.

## Results

Although i.p. pristane injection induces lupus autoantibodies and renal disease in both B6 and BALB/c mice, only B6 mice develop DAH (*Reeves et al., 2009*). MO causes peritoneal inflammation in both

strains, but lupus does not develop. Pristane-induced DAH is abolished by depleting monocytes and Mɸ with clodronate liposomes, whereas neutrophil depletion has little effect (*Zhuang et al., 2017*), suggesting that it is mediated at least partly by monocytes and/or Mɸ. There is little information about monocytes in pristane-treated mice and their relationship to disease pathogenesis. Pristane-induced Ly6C$^{hi}$ monocyte recruitment from the BM to the peritoneum is Ccr2 dependent (*Lee et al., 2009*). A population of highly phagocytic peritoneal Ly6C$^-$CD138$^+$ Mɸ with an anti-inflammatory phenotype is more abundant in MO- vs. pristane-treated mice, whereas Ly6C$^{hi}$CD138$^-$ Mɸ are more abundant in pristane-treated mice (*Han et al., 2017*). We examined the origin of these peritoneal Ly6C$^-$CD138$^+$ Mɸ.

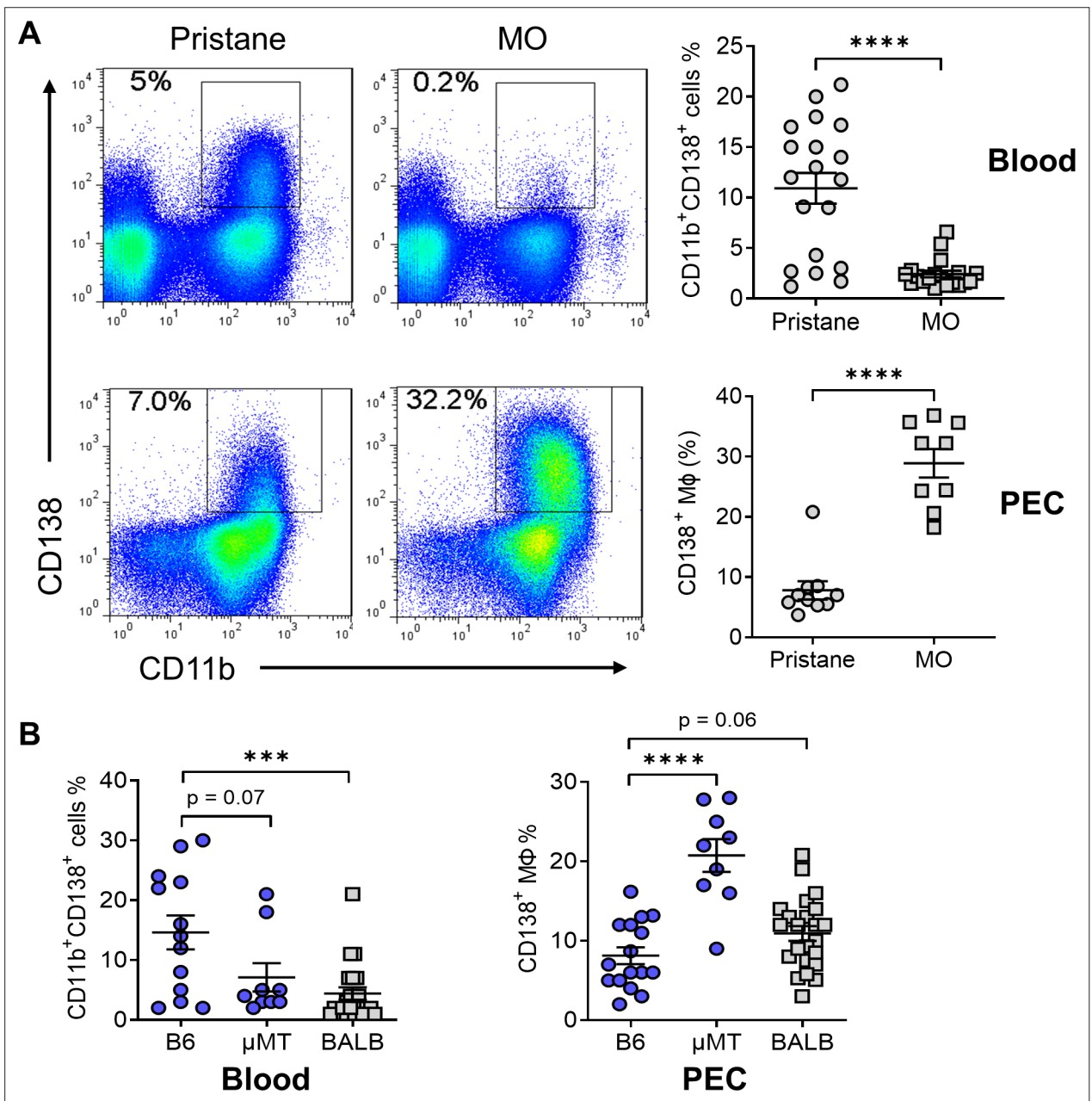

**Figure 1.** CD138$^+$CD11b$^+$ cells in the circulation and peritoneum. (**A**) B6 mice were injected with either pristane or mineral oil (MO). CD138 and CD11b surface expression on peripheral blood leukocytes and peritoneal exudate cells (PECs) was analyzed by flow cytometry at day 14. (**B**) Comparison of the percentages of CD138$^+$ cells in blood and PECs from B6, B6 μMT (B cell deficient), and BALB/c mice. ***p < 0.001; ****p < 0.0001 (Student's *t*-test).

## Circulating CD138+ monocytes

Ly6C[hi] monocytes recruited to the inflamed peritoneum become Ly6C[hi] Mϕ, which subsequently can downregulate Ly6C and develop into Ly6C[−/lo] Mϕ (**Lee et al., 2009**). Ly6C[hi] monocytes also give rise to Ly6C[−/lo] monocytes in the circulation (**Sunderkötter et al., 2004**; **Mildner et al., 2017**), raising the question of whether peritoneal Ly6C[−/lo]CD138+ Mϕ are derived from circulating Ly6C[−/lo] monocytes or from peritoneal Ly6C[hi] Mϕ. CD138+CD11b+ cells were found in the blood of both pristane- and MO-treated B6 mice at 14 days, but were considerably more abundant in pristane-treated mice (**Figure 1A**). In contrast, peritoneal CD138+CD11b+ Mϕ were more abundant in MO-treated mice than pristane-treated mice. BALB/c mice and B6 mice lacking B cells (μMT) do not develop pristane-induced DAH (**Zhuang et al., 2017**; **Barker et al., 2011**). Like MO-treated B6 mice, both B6 μMT and BALB/c mice generally had fewer circulating CD138+CD11b+ cells and more peritoneal CD138+CD11b+ Mϕ than wild-type B6 mice (**Figure 1B**). These observations suggested that circulating CD138+CD11b+ myeloid cells were recruited to the peritoneum and developed into CD138+CD11b+ Mϕ or alternatively, that CD138+CD11b+ Mϕ were generated locally in the peritoneum. As an initial step to distinguish between these possibilities, we carried out more extensive phenotyping of the circulating and peritoneal populations.

Flow cytometry revealed three subsets of CD11b+Ly6G− cells from pristane-treated B6 mice in both the blood and the peritoneum (**Figure 2A**): CD11b+Ly6C[hi]CD138− (R1); CD11b+Ly6C[−/lo]CD138− (R2); and CD11b+Ly6C[−/lo]CD138+ (R3). Ly6C staining was higher on the peritoneal R2/R3 cells (Ly6C[lo]) vs. blood R2/R3 (Ly6C−) cells. In the blood of pristane-treated mice, all three subsets were CD43+ and TremL4+, consistent with a monocyte phenotype (**Figure 2B,C**). The blood R3 and R2 subsets expressed higher levels of CD43 and TremL4 than R1; R3 expressed higher levels than R2. In the blood CD115 was expressed at low levels, mainly on R3 cells (**Figure 2D**). Interestingly, CD43, TremL4, and CD115 were highly expressed by the peritoneal R3, and to a lesser extent R2, subsets from MO-treated, but not pristane-treated, mice (**Figure 2B–D**). As expected, surface staining for CD62L, which is expressed by Ly6C[hi] monocytes that have recently migrated out of the BM and not on Ly6C[lo] monocytes (**Jakubzick et al., 2017**), was stronger in R1 vs. R3 or R2 (**Figure 2E**). Expression was highest in the circulating R1 subset. As classical monocytes are CD115+Ly6C+CD43[lo] whereas non-classical monocytes are CD115+Ly6C−CD43[hi]TremL4+ (**Jakubzick et al., 2017**), we concluded that the circulating R1 cells were classical (Ly6C[hi]) monocytes and the circulating R2 and R3 cells were subsets of non-classical monocytes. Circulating R3 cells also were CX3CR1+CCR2− (**Figure 2F,G**).

Murine non-classical monocytes differentiate in the circulation from Ly6C[hi] precursors, have a Ly6C−CD43+CX3CR1[hi] phenotype and may represent terminally differentiated blood-resident Mϕ rather than true monocytes (**Ginhoux and Jung, 2014**). Consistent with that possibility, circulating R3 monocytes from pristane-treated mice expressed higher levels of the Mϕ marker F4/80 than R1 or R2 monocytes (**Figure 2H**). However, another Mϕ marker (CD64, FcγR1a) was expressed at lower levels on R3 (and R2) vs. R1 (**Figure 2I**). Peritoneal R3 Mϕ stained more intensely for F4/80, CD64, CCR2, and CX3CR1 than circulating R3 monocytes, and with the exception of CX3CR1, exhibited stronger staining than peritoneal R2 Mϕ. Together the data suggest that blood of pristane-treated mice contains classical monocytes along with two subtypes of non-classical monocytes distinguished by the presence (R3) or absence (R2) of CD138 staining. Peritoneal Mϕ also consisted of R1, R2, and R3 populations. Circulating R3 cells had a phenotype compatible with that of 'patrolling' monocytes but also had higher F4/80 staining than circulating R1 or R2 cells, consistent with early Mϕ differentiation. To further characterize the origin of this unusual monocyte subset, we examined CD138 monocyte development in *Ccr2−/−* and *Nr4a1−/−* mice.

## CD138+ monocytes in Ccr2−/− mice

Monocyte egress from the BM is defective in *Ccr2*-deficient mice (**Serbina and Pamer, 2006**) and Ccr2 is necessary for Ly6C[hi] monocytes to migrate to the peritoneum in pristane-treated mice (**Lee et al., 2009**). After Ly6C[hi] (R1) monocytes migrate to the peritoneum, they can downregulate Ly6C. It is not known whether both Ly6C[lo] peritoneal subsets (CD138+, R3 and CD138−, R2) are generated from Ly6C[hi] precursors or if the peritoneal R3 subset is derived from circulating CD138+ monocytes. Circulating Ly6C[hi] (R1) monocytes were strongly Ccr2+, whereas Ly6C−CD138− (R2) and Ly6C−CD138+ (R3) monocytes expressed low levels (**Figure 2F**). In contrast, Ccr2 was expressed by all peritoneal subsets in pristane-treated mice, most strongly by R3. Ccr2 also was expressed by R1, R2, and R3

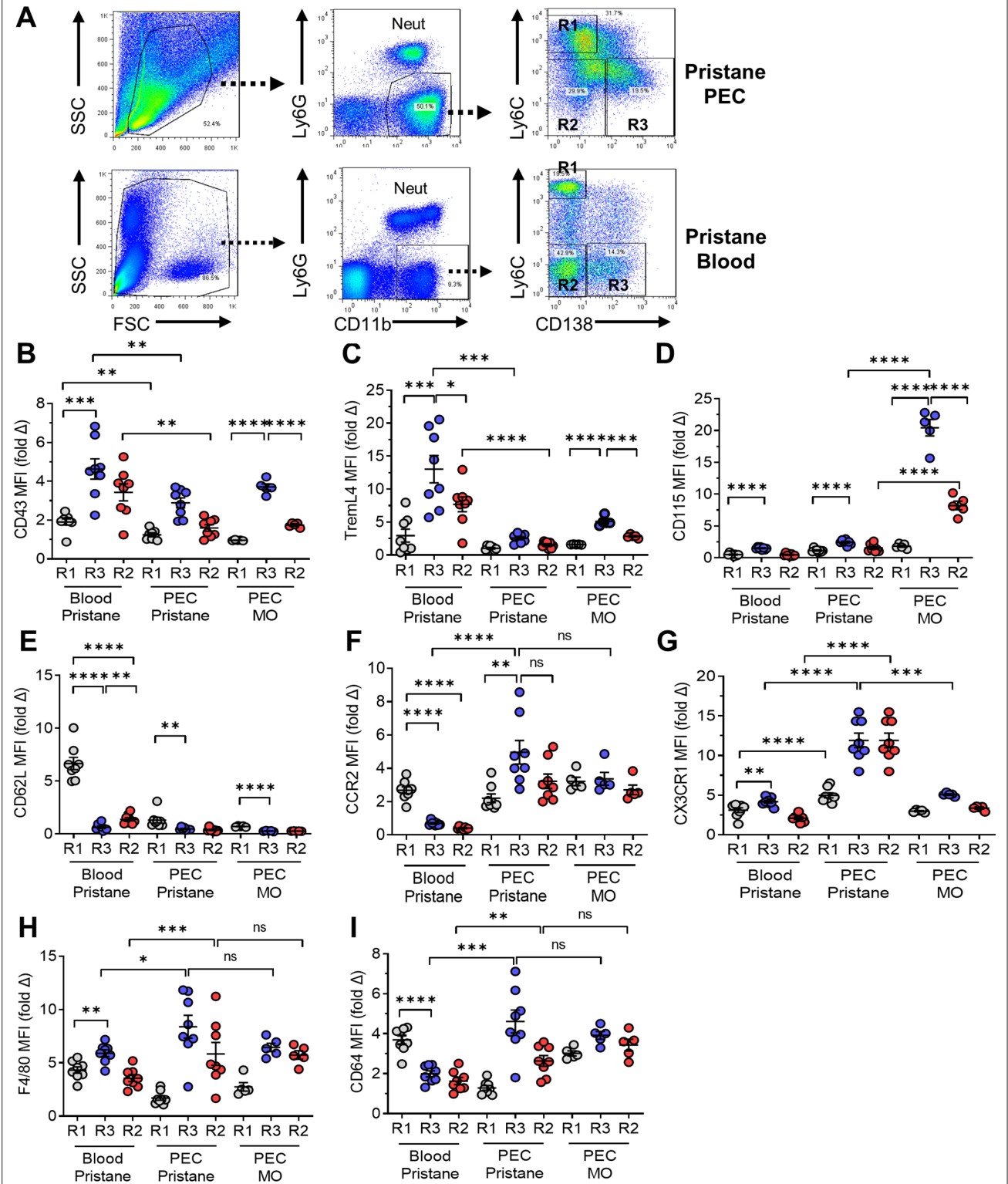

**Figure 2.** Flow cytometry of monocytes surface markers. Peripheral blood mononuclear cells and peritoneal exudate cells (PECs) from pristane- or mineral oil (MO)-treated mice were stained with fluorescently labeled anti-CD11b, Ly6G, Ly6C, and CD138 antibodies. Mean fluorescence intensity (MFI) of CD43, TREML4, CD115, and CD62L was determined on cell subsets. (**A**) Gating strategy for analysis of CD11b⁺Ly6G⁻ PEC and circulating mononuclear cells (blood). R1, CD11b⁺Ly6G⁻Ly6ChiCD138⁻ cells; R2, CD11b⁺Ly6G⁻Ly6C⁻CD138⁻ cells; R3, CD11b⁺Ly6G⁻Ly6C⁻CD138⁺ cells. Staining intensity of CD43 (**B**), TREML4 (**C**), CD115 (**D**), CD62L (**E**), CCR2 (**F**), CX3CR1 (**G**), F4/80 (**H**), and CD64 (**I**) in the R1, R2, and R3 subsets from peripheral blood or PEC from pristane- or MO-treated mice. *p < 0.05; **p < 0.01; ***p < 0.001; ****p < 0.0001 (Student's t-test); ns, not significant.

from MO-treated mice, but at lower levels (*Figure 2F*). Staining of circulating monocytes for Cx3Cr1, which is expressed at high levels on Ly6C⁻ monocytes but is involved in homing of both Ly6Cʰⁱ and Ly6C⁻ monocytes (*Tacke et al., 2007*; *Landsman et al., 2009*), was weak in pristane-treated mice, whereas peritoneal cells (both R2 and R3) expressed higher levels (*Figure 2G*). Expression was higher on peritoneal Mϕ from pristane- vs. MO-treated mice.

Circulating monocyte subsets varied with time after pristane treatment (*Figure 3A*). Ly6Cʰⁱ monocytes (R1) increased first, about 5 days after pristane or MO treatment, while at the same time, the percentage of Ly6C⁻CD138⁻ monocytes (R2) decreased. Circulating Ly6C⁻CD138⁺ monocytes (R3) appeared in pristane-treated mice at day 9 but only small numbers were seen in MO-treated mice (*Figure 3A*).

As expected, circulating R1 monocytes were greatly reduced (but not absent) in pristane-treated *Ccr2−/−* mice (*Figure 3B*). R3 monocytes also were greatly reduced in *Ccr2−/−* mice, suggesting that, like R1 monocytes, their migration from the BM to the circulation was Ccr2 dependent. In contrast to R1 and R3, the percentages of circulating R2 cells were similar in *Ccr2−/−* vs. wild-type mice. In PECs, R1 and R3 cells were absent and R2 cells were present at lower levels in *Ccr2−/−* mice vs. wild-type (*Figure 3B*). Thus, the egress of R1 (Ly6Cʰⁱ) monocytes from the BM and the appearance of R3 (Ly6C⁻CD138⁺) monocytes in the circulation was Ccr2 dependent, whereas the circulating R2 monocyte subset (Ly6C⁻CD138⁻) was unaffected by the absence of Ccr2. R1 and R3 monocytes appeared in the circulation with different kinetics and all three peritoneal Mϕ subsets were reduced in *Ccr2−/−* mice, consistent with the possibility that they were derived from CCR2⁺ R1 monocytes.

## Nr4a1−/− mice lack CD138⁺ monocytes but not CD138⁺ peritoneal Mϕ

Nr4a1 (Nur77) controls the differentiation of Ly6C⁻ monocytes (*Hanna et al., 2011*) and regulates inflammatory responses in Mϕ (*Hanna et al., 2012*). Peripheral blood cells and PECs from wild-type and *Nr4a1−/−* mice were analyzed by flow cytometry 14 days after pristane or MO treatment, gating on forward/side scatter and surface staining characteristics (*Figure 4A*). As expected (*Han et al., 2017*), most peritoneal CD11b⁺Ly6G⁻ cells in MO-treated mice were Ly6Cˡᵒ and many were CD138⁺ (*Figure 4A*). After pristane treatment, percentages of Ly6Cʰⁱ (R1), Ly6CˡᵒCD138⁻ (R2), and Ly6CˡᵒCD138⁺ (R3) peritoneal Mϕ were similar in *Nr4a1−/−* and wild-type mice (*Figure 4A,B*). In contrast, although the percentages of circulating R1 and R2 monocytes were similar, *Nr4a1−/−* mice had a markedly lower percentage of circulating CD138⁺ (R3) monocytes than wild-type mice (*Figure 4C*).

Along with CD11b⁺Tim4⁺ resident (yolk sac-derived) 'large peritoneal macrophages' (LPMs) (*Ghosn et al., 2010*), the uninflamed peritoneum contains CD11b⁺Tim4⁻CD138⁺ cells, which are probably 'small peritoneal macrophages' (SPMs) (*Ghosn et al., 2010*). The percentages of resident LPM in untreated wild-type vs. *Nr4a1−/−* mice were similar, whereas the percentage was higher in *Ccr2−/−* mice (*Figure 4D*), probably reflecting the absence of BM-derived cells. The CD138⁺ SPM subset was increased in *Nr4a1−/−* mice vs. wild-type and *Ccr2−/−* mice, suggesting that they originated from circulating BM-derived (Nr4a1-independent) Ly6C⁺ monocytes rather than Nr4a1-dependent CD138⁺ monocytes.

## Pristane-induced DAH is unaffected by the absence of Nr4a1

Induction of DAH by pristane requires monocytes and/or Mϕ (*Zhuang et al., 2017*). Pristane-treated *Nr4a1−/−* mice developed DAH at a frequency comparable to wild-type controls (*Figure 4E*), suggesting that circulating CD138⁺ (R3) monocytes were not essential for the induction of DAH or were only needed in small numbers. In contrast, DAH was abolished in *Ccr2−/−* mice suggesting that, as in the inflamed peritoneum, migration of Ly6Cʰⁱ monocytes from the BM to the lung is involved in the pathogenesis of DAH.

## Phenotypes of circulating Ly6Cˡᵒ monocytes

Although non-classical monocytes are characteristically Ly6C⁻, CD138 expression has not been reported. We carried out more extensive phenotyping to better define the unusual (CD138⁺) R3 subset. Gating on CD11b⁺Ly6G⁻ (non-neutrophil) peripheral blood cells (mainly monocytes), R2 and R3 could be divided into Ly6Cˡᵒ and Ly6C⁻ subsets. Five groups of circulating myeloid (non-neutrophil) cells could be distinguished: Ly6CʰⁱCD138⁻ (R1), Ly6CˡᵒCD138⁻ (R2A), Ly6C⁻CD138⁻ (R2B), Ly6CˡᵒCD138⁺ (R3A), and Ly6C⁻CD138⁺ (R3B) (*Figure 5A*). All five subsets were present in pristane-treated B6 mice,

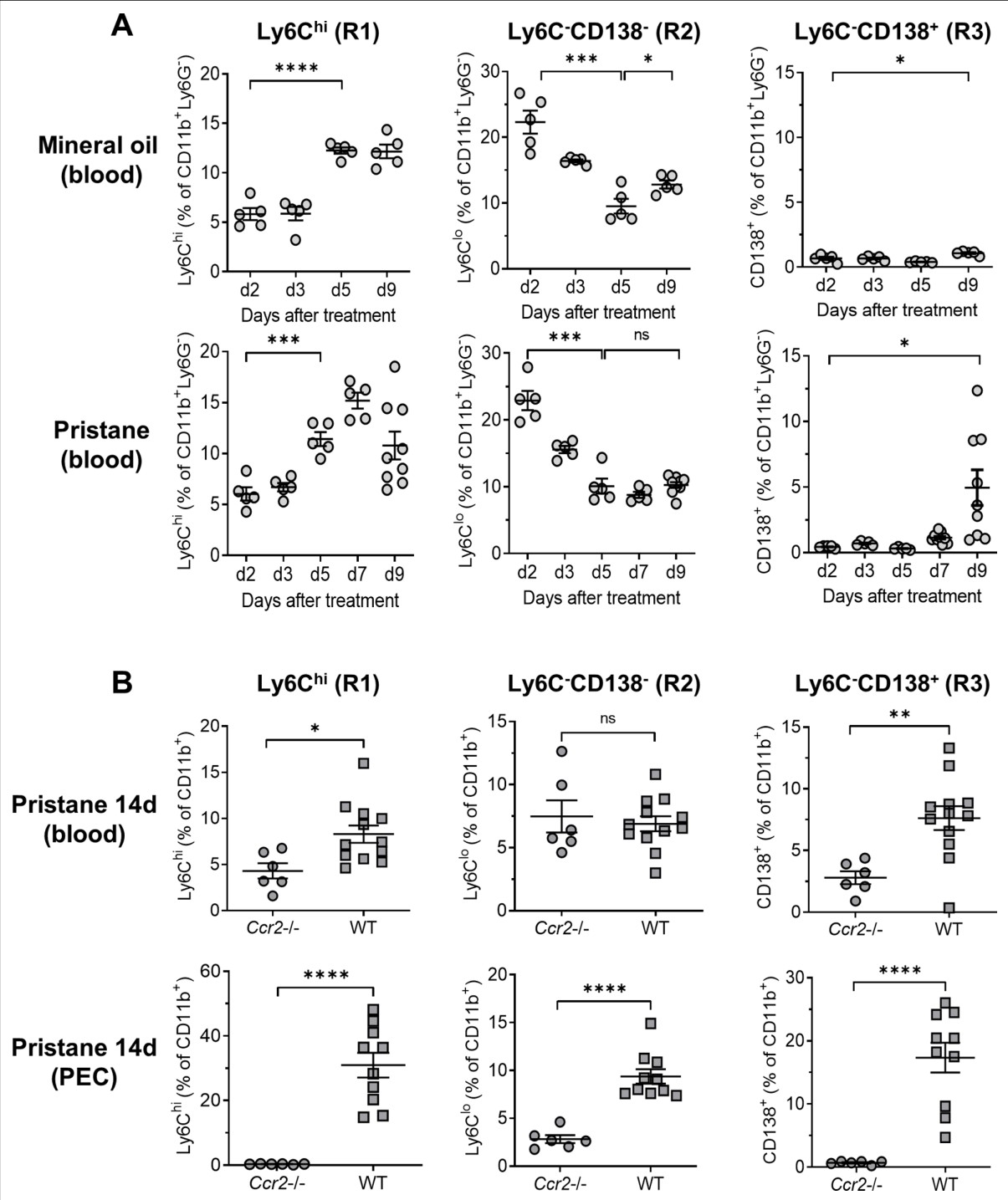

**Figure 3.** Ccr2 dependence of monocyte egress from the bone marrow (BM). B6 mice were treated with either mineral oil or pristane and CD11b⁺Ly6G⁻ myeloid subsets in the blood and peritoneal exudate cell (PEC) were gated as in **Figure 2A**. (**A**) Circulating Ly6Cʰⁱ (R1), Ly6C⁻CD138⁻ (R2) and Ly6C⁻CD138⁺ (R3) monocytes from mineral oil- and pristane-treated mice were assessed at 0–9 days after treatment by flow cytometry. Data are expressed as a percentage of CD11b⁺Ly6G⁻ cells. (**B**) B6 wild-type (WT) and *Ccr2−/−* mice were treated with pristane. R1, R2, and R3 cells as a percentage of CD11b⁺Ly6G⁻ cells were measured in the blood and PEC by flow cytometry 14 days later. *p < 0.05; **p < 0.01; ***p < 0.001; ****p < 0.0001 (Student's *t*-test). ns, not significant.

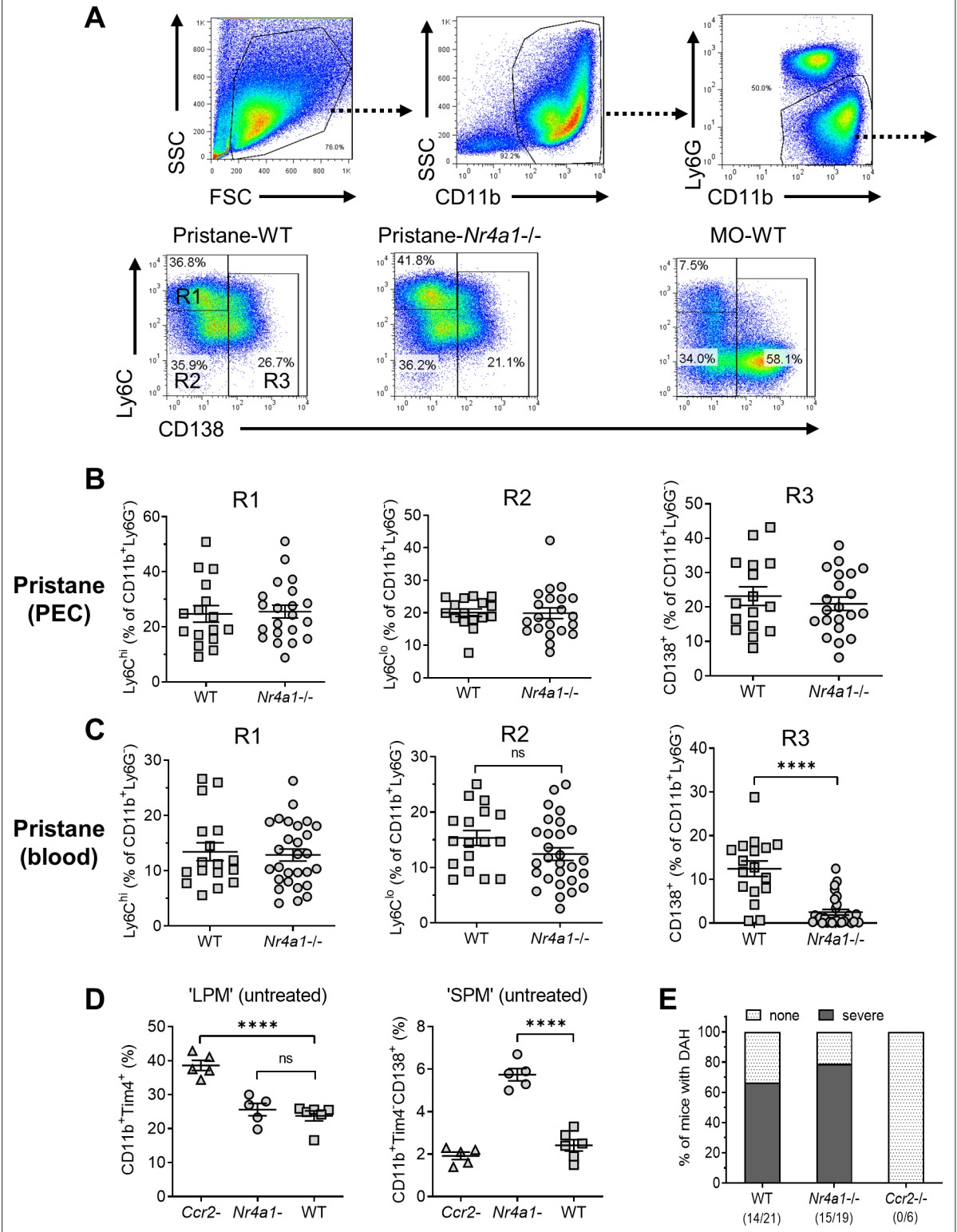

**Figure 4.** Monocyte/Mφ subsets in Nr4a1−/− mice. (**A**) Gating strategy. Peritoneal exudate cells (PECs) from pristane- or mineral oil (MO)-treated wild-type and *Nr4a1*−/− mice were stained with anti-CD11b, Ly6G, Ly6C, and CD138 and analyzed by flow cytometry. PECs were gated on the CD11b+Ly6G− population. R1, Ly6ChiCD138− PECs; R2, Ly6CloCD138− PECs; R3, Ly6CloCD138+ PECs. Gating for circulating (blood) cells was similar. SSC, side scatter; FSC, forward scatter. (**B**) R1, R2, and R3 subsets in the peritoneum (percentage of CD11b+Ly6G− cells). (**C**) R1, R2, and R3 subsets in the peripheral blood (percentage of CD11b+Ly6G− cells). (**D**) Flow cytometry of peritoneal cells in untreated *Ccr2*−/−, *Nr4a1*−/−, and wild-type (WT) mice. *Left*, CD11b+Tim4+

*Figure 4 continued on next page*

*Figure 4 continued*

large peritoneal macrophages (LPMs) as a % of total peritoneal cells. Right, CD11b+Tim4−CD138+ small peritoneal macrophages (SPMs) as a % of total peritoneal cells. (**E**) Frequency of diffuse alveolar hemorrhage (day 14) in pristane-treated WT, *Nr4a1*−/−, and *Ccr2*−/− mice. ****p < 0.0001 (Student's *t*-test). ns, not significant.

whereas only R1, R2A, and R2B were present in phosphate-buffered saline (PBS)- or MO-treated mice. Pristane-treated *Nr4a1*−/− mice showed a pattern similar to that of MO-treated wild-type mice, though R3A and R3B cells were somewhat more abundant (***Figure 5A***). CD138 was not expressed on CD11b+Ly6G− cells from PBS- or MO-treated mice (***Figure 5A***), so Ly6C−/lo cells in pristane-treated mice consisted of R2A, R2B, R3A, and R3B, whereas those from PBS- or MO-treated mice were almost exclusively R2A and R2B (***Figure 5B***). *Nr4a1*−/− mice had low percentages of circulating CD138+ R3A and R3B monocytes vs. wild-type mice, but the percentages were higher than those in MO- or PBS-treated mice (***Figure 5B***, right).

CD138 staining (mean fluorescence intensity, MFI) was much higher in R3A and R3B than in R2A and R2B (***Figure 5C***). CD11b staining intensity showed a similar trend: higher in R3A/R3B vs. R2A/R2B. Since DAH is monocyte dependent and does not develop in pristane-treated mice lacking CR3 [*Itgam*−/− (CD11b) or *Itgb2*−/− (CD18) mice] (***Zhuang et al., 2017***; ***Shi et al., 2014***), we also analyzed circulating monocytes based on Ly6C and CD11b staining. Similar to Ly6C/CD138 staining, five monocyte subsets were defined in pristane-treated mice based on Ly6C and CD11b staining: R4 (Ly6Chi CD11b+/hi), R5A (Ly6Clo CD11b+), R5B (Ly6C− CD11b+), R6A (Ly6Clo CD11bhi), and R6B (Ly6C− CD11bhi) (***Figure 5C*** vs. ***Figure 5D***). CD138 and CD11b staining were both higher on (R6A/R6B) than on R5A/R5B, suggesting that the two subsets of CD138+ monocytes are Ly6Clo CD138+CD11bhi and Ly6C−CD138+CD11bhi, respectively.

Further characterization of circulating monocytes from pristane-treated mice revealed that CD62L was expressed at high levels (as expected) on R1 and R4 cells (***Figure 6A***). CD62L was absent on R3A/R3B and R6A/R6B cells, and was expressed at intermediate levels on R2A/R2B and R5A/R5B (***Figure 6A***). Nr4a1, a transcription factor critical for generating non-classical monocytes (***Hanna et al., 2011***), was expressed intracellularly at high levels in R3A/R3B (and R6A/R6B), low levels in R1 (and R4), and intermediate levels in R2A/R2B (and R5B) (***Figure 6B***). However, levels of Nr4a1 were low in R5A. Ly6C−LFA1hi CCR2lo CX3CR1hi patrolling monocytes adhere and crawl along the endothelial cells in postcapillary venules, a process that is LFA-1 dependent (***Auffray et al., 2007***). Cx3Cl1 (fractalkine), the ligand for Cx3Cr1, is expressed on endothelial cells and mediates adhesion of monocytes to endothelial cells, whereas blocking antibodies to LFA-1 can detach crawling monocytes in *Cx3cr1*+/− mice (***Auffray et al., 2007***). Consistent with the possibility that they are patrolling monocytes, Cx3Cr1 and LFA-1 were expressed more highly by R3A/R3B (and R6A/R6B) monocytes than by the other subsets (***Figure 6C,D***). Expression of CD11c, a differentiation marker expressed by dendritic cells as well as circulating 'foamy monocytes' containing intracellular lipid droplets (***Wu et al., 2009***; ***Xu et al., 2015***), was greatly increased on the R3B (R6B) subset (***Figure 6E***). Phenotypes of the R1, R2A/R2B, and R3A/R3B subsets are summarized in ***Table 1***.

## TLR7 ligand-stimulated TNFα production by Ly6C−CD138+ monocytes

Phenotypic analysis suggested that the Ly6C−CD138+ subset may be a variant of patrolling monocytes, which encounter damaged endothelial cells respond to danger signals via TLR7 and then migrate into tissues where they undergo cMaf/MafB-regulated Mφ differentiation and produce TNFα (***Auffray et al., 2007***; ***Ginhoux and Jung, 2014***). Ly6C− cells are thought to promote focal endothelial necrosis followed by repair. We hypothesized that Ly6C−CD138+ monocytes might respond more aggressively to endothelial injury, potentially exacerbating damage rather than promoting resolution/repair. Along with their acquisition of the dendritic cell marker CD11c (***Figure 6E***), the R3A/R3B (and R6A/R6B) subsets from pristane-treated B6 mice expressed considerably higher levels of TLR7 than other circulating monocyte subsets (***Figure 7A***). These cells also expressed high levels of Treml4 (***Figure 7B***), a protein that amplifies TLR7 signaling (***Ramirez-Ortiz et al., 2015***). As shown in ***Figure 7C***, TLR7 ligand (R848)-stimulated intracellular TNFα staining was comparable in R3 monocytes vs. 'inflammatory' Ly6Chi monocytes. Moreover, TNFα staining was significantly higher in Ly6C−CD138+ (R3) monocytes vs. Ly6C−CD138− (R2) monocytes (***Figure 7C***, right).

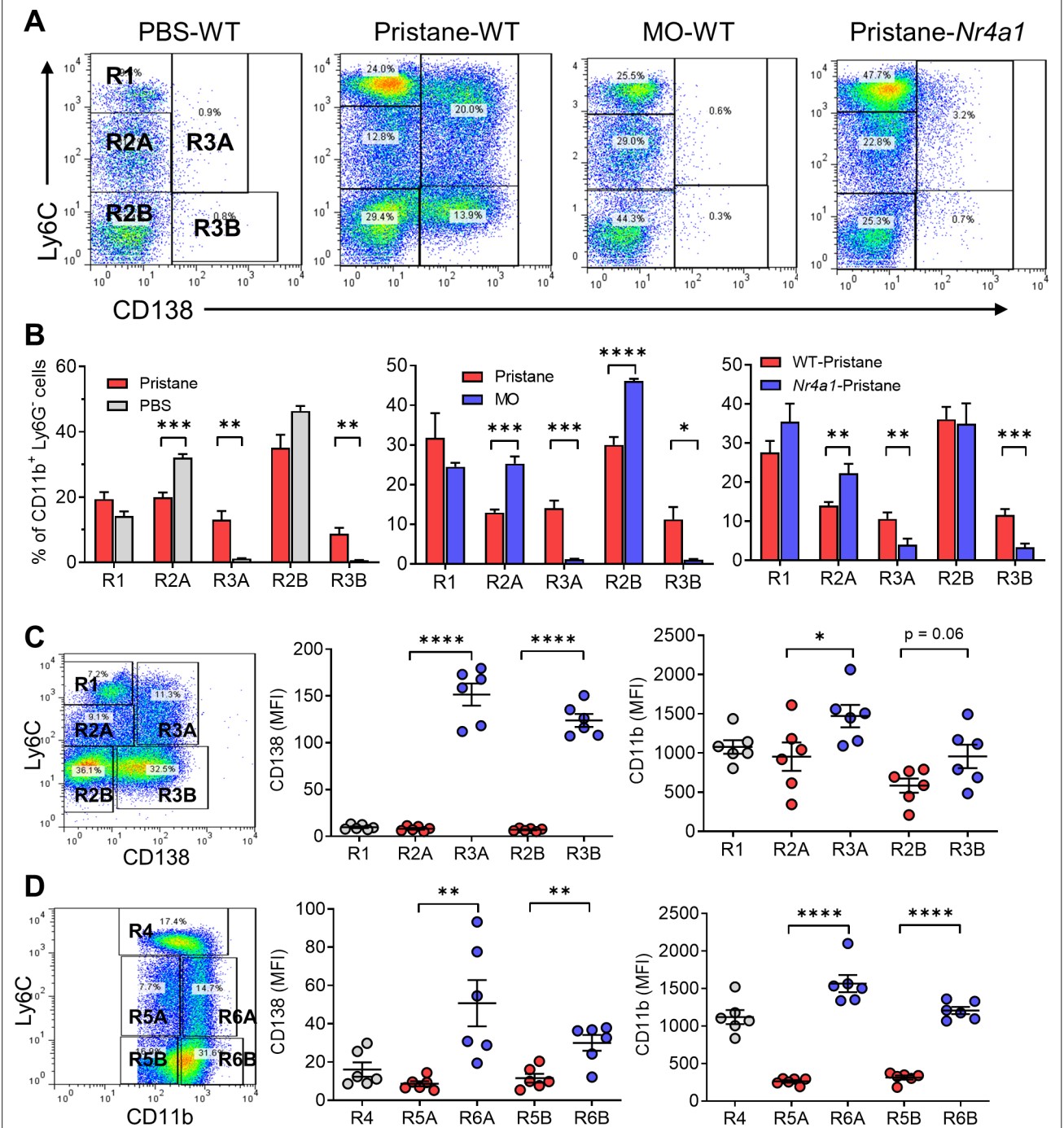

**Figure 5.** Circulating monocyte subsets. (**A**) Gating strategy. Blood cells from pristane- or mineral oil (MO)-treated B6 mice, *Nr4a1*−/− mice, and PBS-treated controls were stained with anti-CD11b, Ly6G, Ly6C, and CD138 and analyzed by flow cytometry, gating on the CD11b+Ly6G− population. R1, Ly6C^hi^CD138−; R2A, Ly6C^lo^CD138−; R2B, Ly6C−CD138−; R3A, Ly6C^lo^CD138+; Ly6C−CD138+. SSC, side scatter; FSC, forward scatter. (**B**) Comparison of circulating CD11b+Ly6G− cells in pristane-treated mice vs. PBS-treated (left) and MO-treated (middle) mice. *Right panel* compares wild-type (WT) vs. *Nr4a1*−/− mice treated with pristane. (**C**) Mean fluorescence intensity (MFI) of CD138 and CD11b staining in the R1, R2A, R2B, R3A, and R3B subsets from pristane-treated mice. (**D**) Alternative gating strategy for circulating CD11b+Ly6G− cells from pristane-treated mice: R4, CD11b+Ly6C^hi^; R5A, Ly6C^lo^CD11b+; R5B, Ly6C−CD11b+; R6A, Ly6C^lo^CD11b^hi^; R6B, Ly6C−CD11b^hi^. Right, CD138 and CD11b staining (MFI) of the R4, R5A/B, and R6A/B subsets. *p < 0.05; **p < 0.01; ***p < 0.001; ****p < 0.0001 (Student's *t*-test).

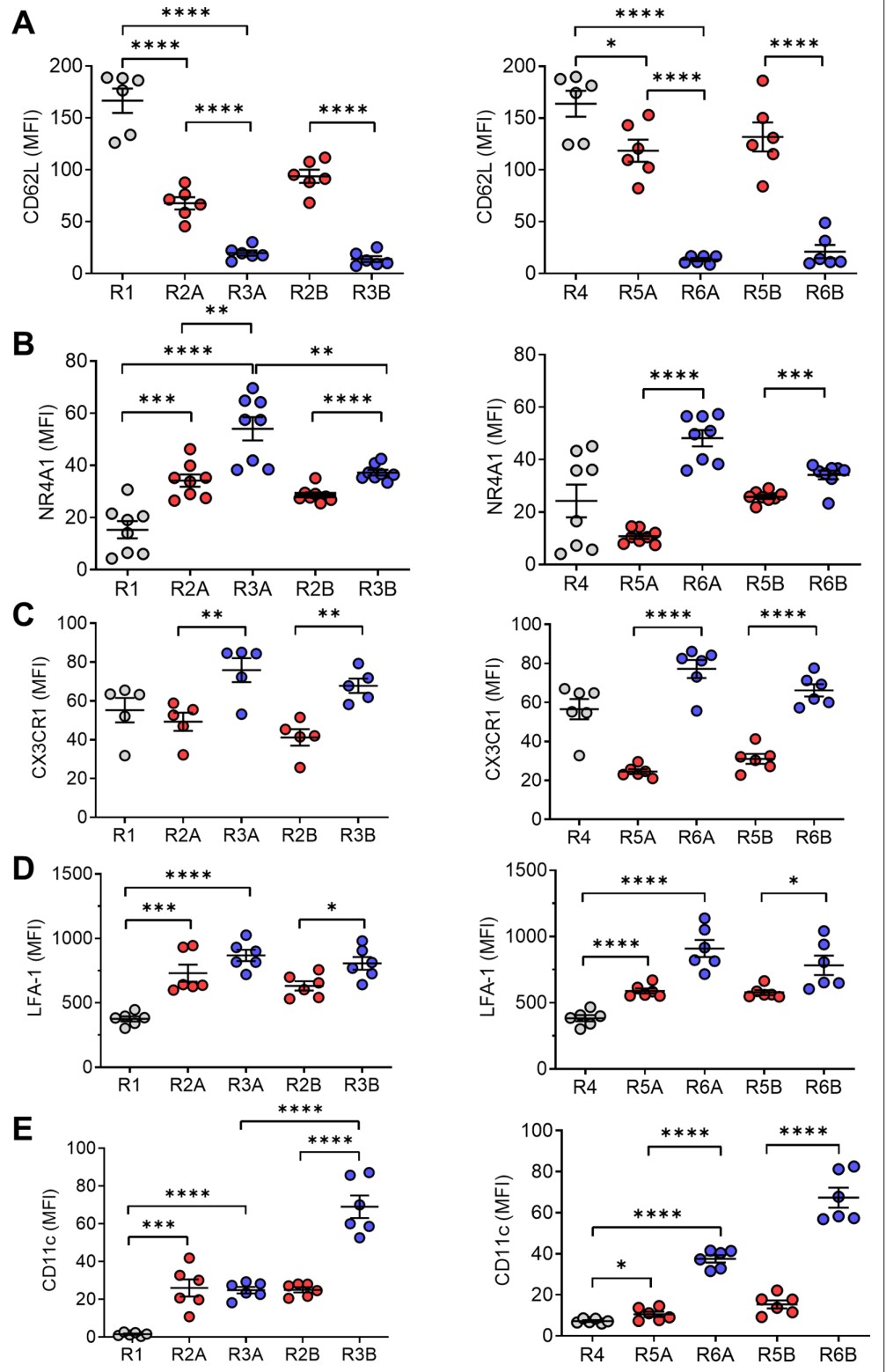

**Figure 6.** Circulating monocyte phenotypes. Circulating monocytes are gated as in *Figure 5* and the staining intensity (MFI) of various markers was ascertained by flow cytometry. (**A**) CD62L, (**B**) Nr4a1, (**C**) Cx3Cr1, (**D**) LFA-1, and (**E**) CD11c. *p < 0.05; **p < 0.01; ***p < 0.001; ****p < 0.0001 (Student's *t*-test).

**Table 1.** Phenotypes of monocyte subsets and pristane- and mineral oil-treated mice.

| | Ly6C | CD138 | CD11b | Nr4a1 | CCR2 | Cx3Cr1 | CD62L | CD11c | LFA1 | Treml4 | TLR7 | TNFa | CD43 | CD115 |
|---|---|---|---|---|---|---|---|---|---|---|---|---|---|---|
| R1 (P)* | ++ | (−) | + | (−) | ++ | + | ++ | (−) | + | (−) | + | ++ | ++ | + |
| R2A (P) | + | (−) | + | + | + | + | + | + | ++ | + | + | ++ | + | + |
| R2B (P) | (−) | (−) | + | + | (−) | + | + | + | ++ | + | + | + | + | + |
| R3A (P) | + | ++ | ++ | ++ | + | ++ | (−) | + | ++ | ++ | ++ | ++ | + | ++ |
| R3B (P) | (−) | ++ | + | + | (−) | ++ | (−) | ++ | ++ | ++ | ++ | + | ++ | ++ |
| R1 (MO)* | ++ | (−) | + | (−) | n.d. | n.d. | n.d. | (−) | n.d. | (−) | + | + | + | n.d. |
| R3A (MO) | + | (−) | ++ | ++ | n.d. | n.d. | n.d. | + | n.d. | ++ | + | ++ | + | n.d. |
| R3B (MO) | (−) | (−) | + | + | n.d. | n.d. | n.d. | ++ | n.d. | ++ | + | + | + | n.d. |

*P, pristane; MO, mineral oil; n.d., not determined.

To investigate whether monocytes from pristane-treated mice were more prone to undergo apoptosis in response to TLR7 signaling, blood from 14-day pristane- or MO-treated mice was incubated for 20 hr with R848 (1 µg/ml) or PBS and then stained with anti-CD11b, Ly6C, and Ly6G antibodies plus PE-conjugated annexin-V. As shown in *Figure 7D* (*left*), the MFI of annexin-V staining was higher on Ly6C$^{hi}$ and Ly6C$^{lo/−}$ monocytes from pristane- vs. MO-treated mice. The MFI of neutrophils from pristane- vs. MO-treated mice was similar. Addition of R848 to the cultures did not significantly change the intensity of annexin-V staining. Histograms revealed that monocyte annexin-V staining was uniformly shifted to the right in pristane- vs. MO-treated mice, but there was only a single peak. In contrast, in neutrophils a second peak with more intense staining, most likely representing apoptotic cells, was seen, more prominently in pristane- vs. MO-treated mice (*Figure 7D*, *right*, *arrows*). Monocytes did not exhibit this second peak. Annexin-V staining also was shifted to the right in freshly isolated monocytes (without in vitro culture) from pristane- vs. MO-treated mice (not shown). Intracellular staining of freshly isolated cells with anti-activated caspase-3 antibodies was negative (not shown), suggesting that the annexin-V$^+$ monocytes from pristane-treated mice were not apoptotic.

## Low TremL4 expression in Nr4a1−/− mice

TremL4 amplifies proinflammatory signaling through TLR7 by regulating the recruitment of MyD88 (*Hemmi et al., 2009*; *Ramirez-Ortiz et al., 2015*; *Nedeva et al., 2020*; *Gonzalez-Cotto et al., 2020*). It is expressed by Ly6C$^−$ monocytes and Nr4a1$^+$Ly6C$^{hi}$ monocytes committed to differentiate into Ly6C$^−$ monocytes (*Briseño et al., 2016*). Circulating CD11b$^+$Treml4$^+$ monocytes increased in pristane-treated wild-type mice, and to a lesser degree in untreated wild-type mice vs. pristane-treated *Nr4a1−/−* mice (*Figure 8A*). In both wild-type and *Nr4a1−/−* mice Treml4 staining intensity was highest in circulating CD138$^+$Ly6C$^−$ monocytes (R3) and was lower in CD138$^−$Ly6C$^{−/lo}$ (R2A/R2B) monocytes (*Figure 8B*). R1 cells were only weakly positive and neutrophils did not stain for TremL4.

Nearly all circulating CD138$^+$ (R3) cells were TremL4$^+$ (*Figure 8C*, boxes). There was a suggestion that Ly6C$^{hi}$ (R1) cells may acquire Treml4 surface staining concomitantly with downregulation of Ly6C and upregulation of CD138 (*Figure 8C*, arrows, lower right). The acquisition of TremL4 and CD138 staining was attenuated in *Nr4a1−/−* mice. In contrast, although signaling through the type I interferon receptor (Ifnar1) blocks the downregulation of Ly6C in peritoneal Mɸ, *Ifnar1−/−* and wild-type mice had similar numbers of circulating CD138$^+$TremL4$^+$ monocytes (*Figure 8C*). Mean Nr4a1 staining intensity in total CD11b$^+$Ly6G$^−$ cells from pristane- and PBS-treated mice correlated with the percentage of circulating TremL4$^+$CD11b$^+$Ly6G$^−$ cells (*Figure 8D*), suggesting that Nr4a1 might regulate TremL4. Consistent with that possibility, pristane and MO treatment both increased the fluorescence intensity of Nr4a1 and TremL4 in circulating CD11b$^+$Ly6G$^−$ cells (*Figure 8E*).

## Ly6C$^−$CD138$^+$ monocytes are associated with DAH

Circulating (Nr4a1-dependent) Ly6C$^−$CD138$^+$ monocytes were present in pristane-treated mice with DAH, but not MO-treated mice without DAH. However, *Nr4a1−/−* mice were susceptible to the induction of DAH by pristane (*Figure 4E*). Circulating Ly6C$^-$CD138$^+$ monocytes were substantially lower in BALB/c (resistant to DAH) vs. B6 (DAH-susceptible) mice and there was a similar trend in

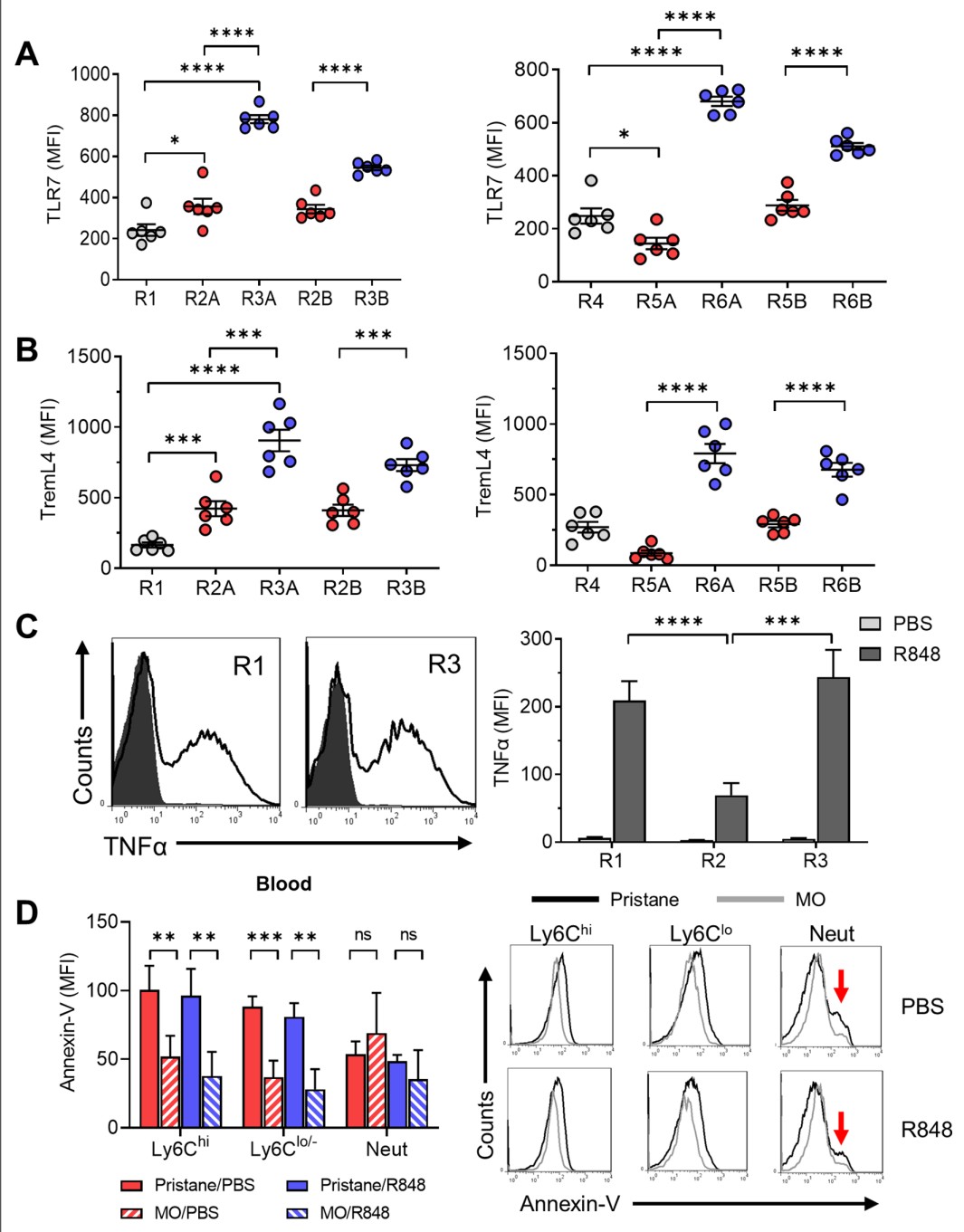

**Figure 7.** Function of CD138+ monocytes. Circulating monocytes from pristane-treated B6 mice were gated as in *Figure 5* and the staining intensity (MFI) of TLR7 (**A**) and TremL4 (**B**) was ascertained by flow cytometry. (**C**) R848-stimulated TNFα production in R1, R2, and R3 monocytes from pristane-treated mice (14 days). Circulating leukocytes were incubated 5 hr with R848 or PBS and then surface stained with anti-CD11b, Ly6G, Ly6C, and CD138 antibodies and intracellularly stained with anti-TNFα. *Left,* histograms of intracellular TNFα staining (R1 and R3 subsets). *Right*, TNFα staining of blood cells cultured 5 hr with R848 or PBS. (**D**) Annexin-V staining (*left*, mean fluorescence intensity; *right*, representative histograms) of circulating CD11b+ monocyte subsets (Ly6Chi Ly6G− and Ly6Clo/−Ly6G−) and CD11b+Ly6G+ neutrophils (Neut) from pristane- or mineral oil (MO)-treated mice cultured for 20 hr with R848 or PBS. Red arrows, annexin-Vhi apoptotic neutrophils. *p < 0.05; **p < 0.01; ***p < 0.001; ****p < 0.0001 (Student's *t*-test); ns, not significant.

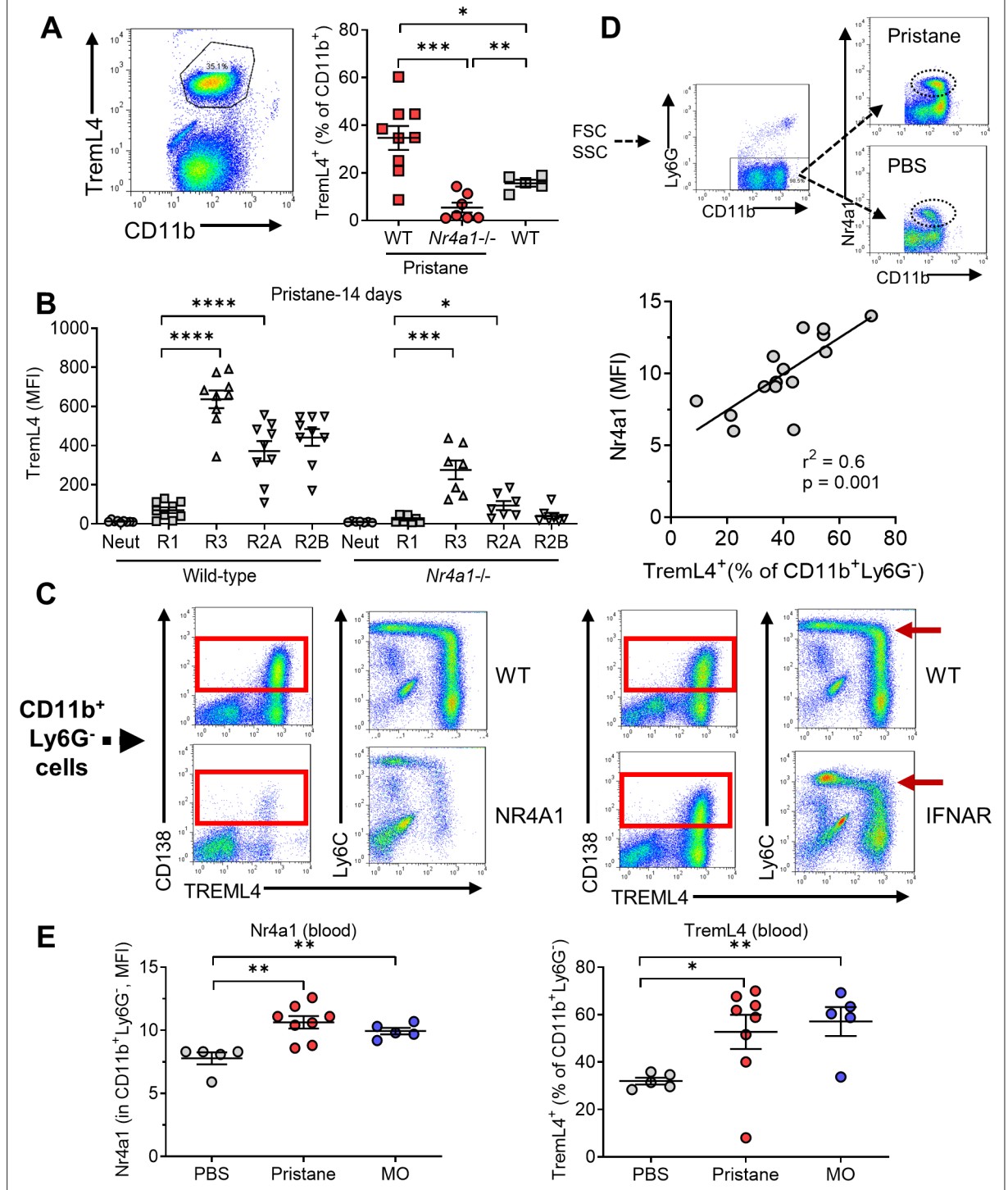

**Figure 8.** Low TremL4 expression in Nr4a1−/− mice. (**A**) Blood cells were stained with anti-CD11b and anti-Treml4 monoclonal antibodies. CD11b⁺Treml4⁺ cells as a percentage of total CD11b⁺ cells were determined by flow cytometry in pristane-treated wild-type (WT) and *Nr4a1−/−* mice and in untreated WT mice. (**B**) Wild-type and *Nr4a1−/−* mice were treated with pristane and CD11b⁺Ly6G⁻ blood cells were gated as in *Figure 2A*. CD11b⁺Ly6G⁺ neutrophils (% of total CD11b⁺ cells) and Ly6C^hi (R1), Ly6C⁻/loCD138⁻ (R2), and Ly6C⁻/loCD138⁺ (R3) monocytes (% of CD11b⁺Ly6G⁻ cells) were determined. (**C**) *Left*, TremL4, CD138, and Ly6C staining of circulating CD11b⁺Ly6G⁻ cells from pristane-treated WT vs. *Nr4a1−/−* mice. *Right*, CD138, and Ly6C staining of circulating CD11b⁺Ly6G⁻ cells from wild-type (WT) vs. *Ifnar1−/−* mice. Nearly all CD138⁺ cells were TremL4⁺ (red boxes). *Red arrow*, Ly6C^hi cells. (**D**) *Top*, gating strategy for identifying Nr4a1⁺ cells. *Bottom*, linear regression analysis of intracellular Nr4a1 and TremL4 staining in total CD11b⁺Ly6G⁻ cells from PBS- plus pristane-treated B6 mice. (**E**) Flow cytometry of Nr4a1 (intracellular staining) and TremL4 (surface staining) in circulating CD11b⁺Ly6G⁻ cells from wild-type mice treated with PBS, pristane, or mineral oil (MO). *p < 0.05; **p < 0.01; ***p < 0.001; ****p < 0.0001 (Student's *t*-test).

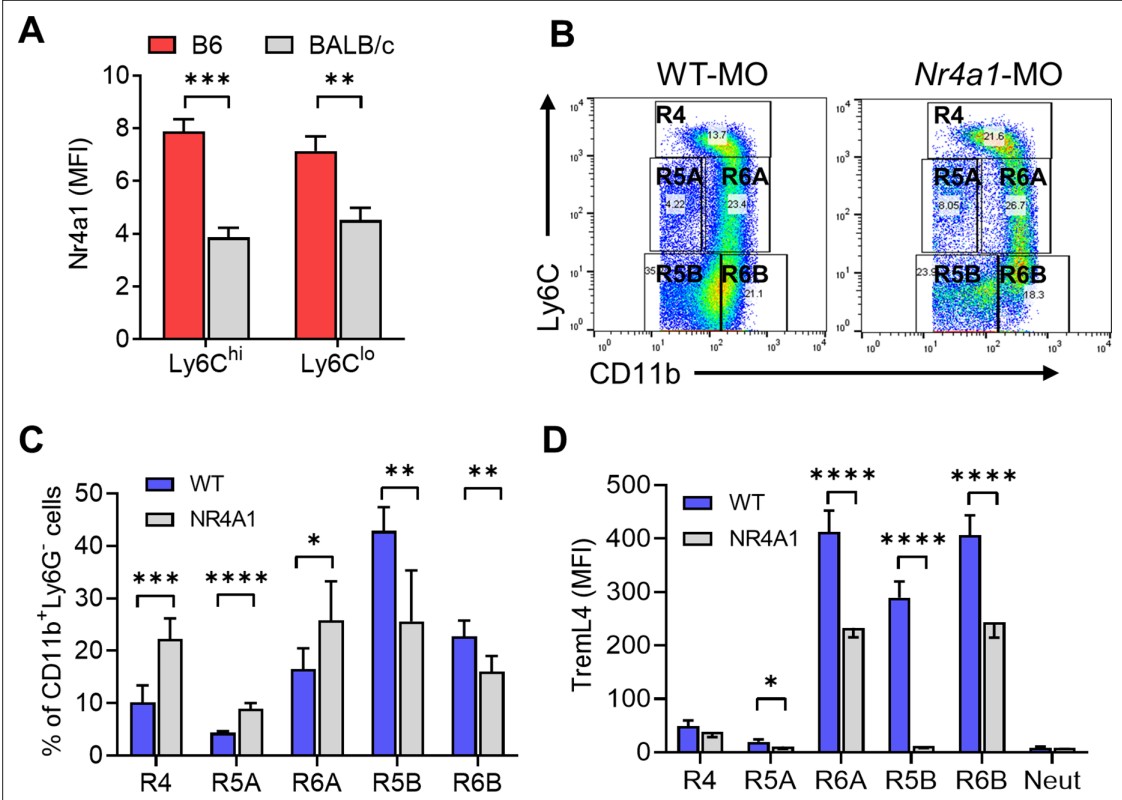

**Figure 9.** Ly6C$^{lo}$CD138$^+$ monocytes are increased in mice that develop diffuse alveolar hemorrhage (DAH). (**A**) Intracellular Nr4a1 staining in the CD11b$^+$Ly6C$^{hi}$ and CD11b$^+$Ly6C$^{-/lo}$ monocytes from untreated B6 and BALB/c mice. (**B**) CD11b and Ly6C staining of circulating Ly6G- cells in mineral oil (MO)-treated mice. Gated as in **Figure 5A**. (**C**) Percentages of R4, R5A, R5B, R6A, and R6B monocytes in wild-type B6 (WT) vs. B6 *Nr4a1–/–* mice treated with MO. (**D**) TremL4 staining of circulating R4, R5A, R5B, R6A, and R6B monocytes and neutrophils (Neut) from MO-treated mice. *p < 0.05; **p < 0.01; ***p < 0.001; ****p < 0.0001 (Student's *t*-test).

pristane-treated B-cell-deficient (μMT) B6 mice, which also fail to develop DAH (***Zhuang et al., 2017***; **Figure 1B**). Intracellular Nr4a1 staining (flow cytometry) was lower in circulating Ly6C$^{hi}$ and Ly6C$^{lo}$ monocytes from untreated BALB/c vs. B6 mice (**Figure 9A**). Thus, increased numbers of circulating, Nr4a1-dependent, Ly6C$^{lo}$CD138$^+$ monocytes and a high level of monocyte Nr4a1 protein expression were associated with susceptibility to DAH.

Although Nr4a1-dependent Ly6C$^-$ patrolling monocytes are present in untreated B6 mice, they did not express CD138 in RNA-Seq studies (***Mildner et al., 2017***). There were few CD138$^+$Ly6C$^-$ monocytes in either untreated or MO-treated mice (**Figures 1 and 5A**). In MO-treated mice, CD138$^-$Ly6C$^-$ monocytes were present in the circulation (**Figure 5A**), but they were not reduced in *Nr4a1–/–* mice (**Figures 4C and 5A**). We hypothesized that MO-treated mice may have Nr4a1-dependent Ly6C$^{-/lo}$CD138$^-$ (R2) monocytes equivalent to the R3 subset in pristane-treated mice. In view of the extremely low number of Ly6C$^{-/lo}$CD138$^+$ cells in MO-treated or *Nr4a1–/–* mice (**Figure 4** and **Figure 5**), we examined Ly6C$^{-/lo}$CD11b$^{hi}$ monocytes (**Figure 5C,D**), which are likely to be similar to the Ly6C$^{-/lo}$CD138$^+$ subset (**Figure 6**). Significant numbers of circulating Ly6C$^{-/lo}$CD11b$^{hi}$ (R6A/R6B) monocytes were seen in MO-treated wild-type mice and *Nr4a1–/–* mice (**Figure 9B,C**). R6A cells were higher in *Nr4a1–/–* mice vs. wild type, whereas R6B cells were significantly lower in *Nr4a1–/–* mice. R5B cells also were reduced in *Nr4a1–/–* mice (**Figure 9C**). Thus, the development in MO-treated mice of Ly6C$^-$CD11b$^{hi}$ (R6B) and Ly6C$^-$CD11b$^+$ (R5B) monocytes, like the development of Ly6C$^-$CD138$^+$ (R3B) monocytes in pristane-treated mice, was Nr4a1 dependent. Together, the data suggest the following: (1) Nr4a1-dependent Ly6C$^-$ monocytes express CD138 in pristane-treated mice that develop DAH, but not in mice that are resistant to pristane-induced DAH (MO-treated B6, B6 μMT, and BALB/c) and (2) the R5B and R6B subsets in MO-treated mice are Nr4a1 regulated.

Surface staining of blood cells from MO-treated mice showed high levels of TremL4 expression in the R5B, R6A, and R6B subsets (*Figure 9D*). The expression of TremL4 was reduced substantially in *Nr4a1*−/− mice, further suggesting that TremL4 might be Nr4a1 regulated. This possibility was examined in a murine Mϕ cell line (RAW264.7).

## Nr4a1 regulates TremL4 expression

The TLR7 ligand R848 activated RAW264.7 cells, resulting in increased CD86, Nr4a1, and TremL4 staining (*Figure 10A*). lipopolysaccharide (LPS), a TLR4 ligand, had a similar effect (not shown). *Nr4a1* mRNA expression increased rapidly after adding R848, peaking at 1 hr (*Figure 10B*). *Treml4* and *Tnfa* mRNA peaked later on at ~3 hr. TremL4 surface staining also was enhanced by culturing RAW264.7 cells with pristane (12.5–200 ng/ml emulsified in bovine serum albumin [BSA]) (*Figure 10C*). The effect was dose dependent and specific for pristane, as culture with MO did not enhance TremL4 staining (*Figure 10D*).

To further examine Nr4a1 regulation of *Treml4* expression, RAW264.7 cells were cultured with LPS in the presence of *Nr4a1* or control siRNA. As expected, *Nr4a1* siRNA downregulated LPS-stimulated *Nr4a1* mRNA expression at 1 hr (*Figure 10E*). *Nr4a1* siRNA also downregulated LPS-stimulated *Treml4* mRNA expression at 3 hr. Adherent peritoneal Mϕ from untreated mice behaved similarly. As expected, *Nr4a1* expression was minimal in PBS-treated adherent peritoneal Mϕ from *Nr4a1*−/− mice compared with wild-type controls (*Figure 10F*, left). Unexpectedly, in adherent peritoneal cells of *Nr4a1*−/− mice, *Nr4a1* increased slightly after LPS stimulation, probably due to the expression of a truncated fragment of the *Nr4a1* gene encoding part of the N-terminal domain in B6.129S2-*Nr4a1*[tm1Jmi]/J (*Nr4a1*−/−) mice (*Koenis et al., 2018*). TremL4 expression was lower in PBS-treated adherent peritoneal cells from *Nr4a1*−/− mice vs. wild-type controls (*Figure 10F*, right). *Nr4a1* as well as *Treml4* expression was significantly higher in both PBS- and LPS-treated adherent cells from wild-type mice vs. *Nr4a1*−/− mice (*Figure 10F*).

## Discussion

Transcriptional profiling of circulating monocytes from uninflamed B6 mice shows a single population of Nr4a1-regulated Ly6C− monocytes (*Mildner et al., 2017*). These cells mostly remain within the blood vessels where they patrol the vascular endothelium and promote TLR7-dependent repair of damaged endothelial cells (*Auffray et al., 2007*; *Carlin et al., 2013*). Previous studies have not fully addressed whether inflammation alters Ly6C− monocytes. We show that Ly6C− monocytes are more heterogeneous during inflammation than in the resting state. Pristane-induced lupus with DAH and small vessel vasculitis is a monocyte/Mϕ-dependent disorder involving lung microvascular endothelial injury (*Zhuang et al., 2017*; H Zhuang et al., *In Revision*). Like pristane, MO causes a chronic inflammatory response, but it is not associated with DAH, pulmonary vasculitis, or endothelial damage. We found that Nr4a1-regulated patrolling (Ly6C−) monocytes had different phenotypes in mice treated with pristane (CD138+) vs. MO (CD138−). Pristane directly affected cells of the monocyte/Mϕ lineage, increasing the expression of TremL4, a protein that amplifies TLR7 signaling and altering the normal plasma membrane asymmetry, as shown by increased annexin-V binding to phosphatidylserine on the plasma membrane's outer leaflet. Ly6C− monocytes from pristane-treated mice produced large amounts of TNFα in response to TLR7 ligand, but despite their high annexin-V staining, did not appear to be more susceptible to apoptosis than Ly6C− monocytes from MO-treated mice. The data suggest that pristane alters the activation state of Ly6C− patrolling monocytes, potentially influencing whether endothelial damage persists or is repaired.

### Phenotypes of Ly6C− monocytes in pristane vs. MO-treated mice

Ly6C− monocytes developing from BM-derived Ly6C[hi] precursors are thought to be a single-cell population (*Sunderkötter et al., 2004*; *Mildner et al., 2017*) that monitors the vascular endothelium and promotes the TLR7-dependent removal of damaged cells (*Auffray et al., 2007*; *Carlin et al., 2013*). Development of this lineage requires Nr4a1 (*Hanna et al., 2011*) and C/EBPβ (*Mildner et al., 2017*). Circulating monocytes in pristane-treated mice included classical Ly6C[hi] inflammatory monocytes (R1) and two subsets of Ly6C− monocytes distinguished by the expression of CD138 and CD11b: Nr4a1-independent CD138−CD11b+ monocytes (R2/R5) and Nr4a1-regulated CD138+CD11b[hi] monocytes

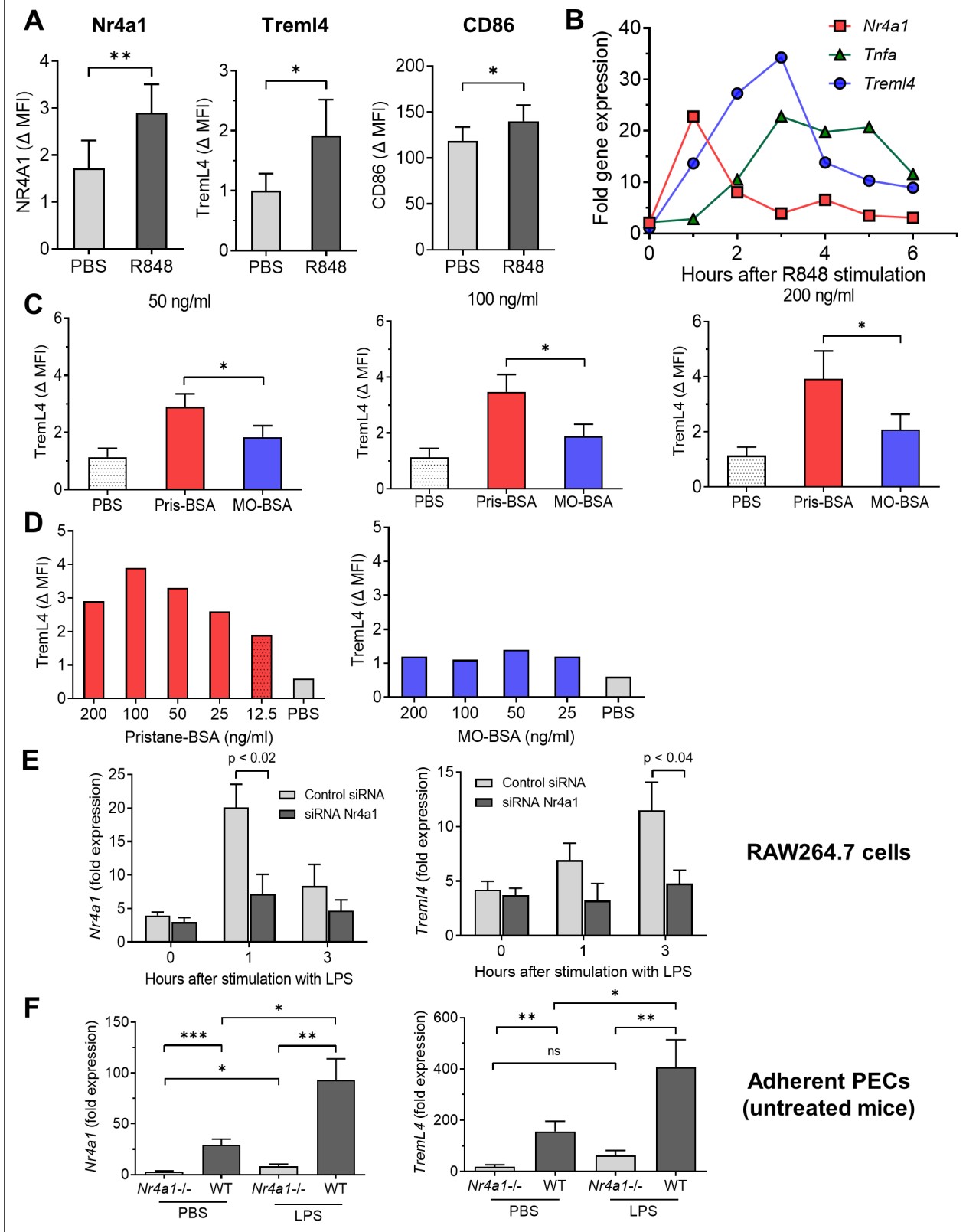

**Figure 10.** Nr4a1 regulates TremL4. (**A**) R848 activated expression (24 hr) of Nr4a1, TremL4, and CD86 proteins in RAW264.7 cells (flow cytometry; MFI, mean fluorescence intensity). (**B**) R848-stimulated mRNA expression of *Nr4a1*, *Tnfa*, and *TremL4 (qPCR)*. (**C, D**) TremL4 expression (MFI) in RAW264.7 cells by flow cytometry after culture with pristane or mineral oil (MO) 12.5–200 ng/ml emulsfied in bovine serum albumin (BSA) or with PBS for 24 hr. (**E**) *Nr4a1* and *Treml4* expression (qPCR) in LPS-stimulated RAW264.7 cells cultured with Nr4a1 or control siRNA. (**F**) *Nr4a1* and *Treml4* expression (qPCR) in

*Figure 10 continued on next page*

*Figure 10 continued*

adherent peritoneal exudate cells (PECs) from untreated wild-type (WT) or *Nr4a1−/−* mice treated for 3 hr with LPS or PBS. *p < 0.05; **p < 0.01; ***p < 0.001 (Student's *t*-test).

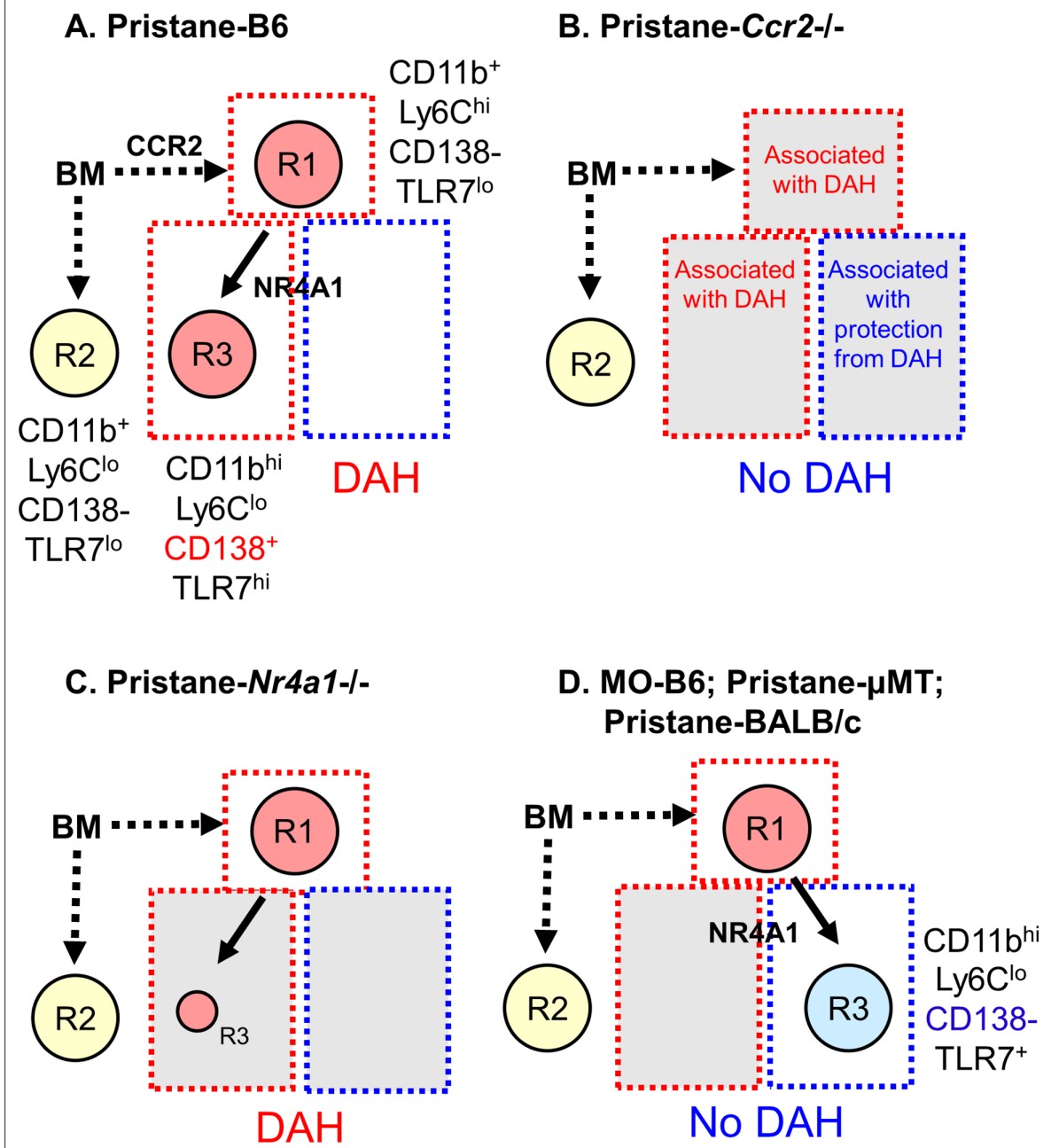

**Figure 11.** Monocyte subsets in pristane- and mineral oil (MO)-treated mice. (**A**) Pristane-treated B6 mice; (**B**) pristane-treated B6 *Ccr2−/−* mice; (**C**) pristane-treated B6 *Nr4a1−/−* mice; (**D**) MO-treated B6 mice, pristane-treated B6 µMT mice, and pristane-treated BALB/c mice. Phenotypes of bone marrow (BM)-derived monocyte subsets R1, R2, and R3 and their dependence on CCR2 and Nr4a1 are indicated. Subsets that are associated with susceptibility to diffuse alveolar hemorrhage (DAH) are highlighted in red. Subsets associated with resistance to DAH are highlighted blue. Subsets that have no apparent relationship with DAH are highlighted yellow. R3 monocytes are Nr4a1 dependent, but have somewhat different phenotypes in pristane- (**A**) vs. MO-treated (**B**) mice. In pristane-treated mice, they appear to be permissive for induction of DAH, whereas in MO-treated mice, they may inhibit the development of DAH.

(R3/R6) (*Figure 11*). The Nr4a1-independent R2/R5 subset was strongly positive for the monocyte markers CD43 and Treml4 (*Figure 2*), suggesting that these Ly6C$^{lo}$ cells are monocytes. Nr4a1-independent Ly6C$^{-}$ monocytes have been encountered before but their function is unclear (*Carlin et al., 2013*).

Ly6C$^{-}$CD138$^{-}$ (R2B) Ly6C$^{-}$CD138$^{+}$ (R3B) monocytes were clearly distinguishable by flow cytometry in pristane-treated mice (*Figure 5A, C*). Consistent with the presence of cells expressing intermediate levels of Ly6C in untreated mice (*Mildner et al., 2017*), pristane-treated mice also had circulating Ly6C$^{lo}$ cells that were CD138$^{-}$ (R2A) and CD138$^{+}$ (R3A) (*Figure 5A, C*). Analogous populations corresponding to R2A/R2B and R3A/R3B were defined by Ly6C and CD11b staining (*Figure 5D*). The differential requirement for Nr4a1 (*Figures 4C and 5B*) indicates R2B and R3B are distinct subtypes of Ly6C$^{-}$ monocytes. Consistent with a BM-derived Ly6C$^{hi}$ precursor (*Yona et al., 2013*; *Sunderkötter et al., 2004*; *Mildner et al., 2017*; *Hettinger et al., 2013*), R3 monocytes were greatly reduced in pristane-treated *Ccr2*−/− mice (*Figure 3B*). In pristane-treated mice, R2 monocytes were unaffected by Ccr2 deficiency and therefore unrelated to R3 monocytes.

Between days 5 and 9 after pristane treatment, circulating R3 monocytes increased significantly while R2 was unchanged (*Figure 3A*). In MO-treated mice, the opposite pattern was seen: R3 monocytes remained unchanged, whereas R2 increased from days 5 to 9. In pristane-treated *Nr4a1*−/− mice, the Ly6C$^{-/lo}$CD138$^{+}$ (R3A/R3B) populations were substantially lower than in wild-type controls, but higher than in PBS- or MO-treated mice (*Figure 5B* and *Figure 11*). Due to the low expression of CD138 on monocytes from MO-treated mice, we used an alternative gating strategy to identify subsets of Ly6C$^{-/lo}$ cells. Ly6C$^{-}$CD11b$^{hi}$ (R6) and Ly6C$^{-}$CD11b$^{+}$ (R5) monocytes were phenotypically similar to R3 and R2, respectively (*Figure 5D* and *Figure 6*). Ly6C$^{-}$ cells (R5B, R6B) in MO-treated mice were Nr4a1 dependent (*Figure 9B,C*), suggesting that in MO-treated mice, the Nr4a1-dependent Ly6C$^{-}$CD11b$^{hi}$ subset was CD138−, whereas in pristane-treated mice it was largely CD138+ (*Figure 11*). Thus, CD138 staining of Ly6C$^{-}$ monocytes was related to the type of inflammatory stimulus (pristane vs. MO).

## Monocytes in mice with pristane-induced DAH

Pristane-induced lupus in B6 mice is complicated by severe lung disease closely resembling SLE-associated DAH with pulmonary vasculitis in humans (*Zhuang et al., 2017*; *Zamora et al., 1997*). DAH and ANCA-negative small vessel vasculitis are abolished by depleting monocytes/Mφ with clodronate liposomes and are absent in mice lacking immunoglobulin or C3, suggesting the involvement of immune complexes (*Zhuang et al., 2017*). DAH in SLE patients also shows evidence of an immune complex pathogenesis (*Hughson et al., 2001*; *Al-Adhoubi and Bystrom, 2020*). DAH in pristane-induced lupus is associated with lung microvascular injury (*Zhuang et al., 2017*, in revision) and monocyte recruitment (*Lee et al., 2019*). We confirmed the importance of Ccr2 in DAH (*Figure 4E*), underscoring the central role of Ly6C$^{hi}$ monocyte recruitment (*Lee et al., 2019*). R2 monocytes were unaffected by the absence of Ccr2, suggesting that they were dispensable for the induction of DAH (*Figure 3B* and *Figure 11*). In contrast, R1 monocytes (deficient but not absent in *Ccr2*−/− mice) appeared to be required. R3 monocytes also were deficient in *Ccr2*−/− mice (*Figure 3B*), but are derived from Ly6C$^{hi}$ precursors (*Sunderkötter et al., 2004*; *Mildner et al., 2017*). *Nr4a1*−/− mice had normal numbers of R1 monocytes and were susceptible to pristane-induced DAH (*Figure 4C, E* and *Figure 5A*). R3 monocytes were greatly reduced (but not absent) in *Nr4a1*−/− mice (*Figure 4C*), suggesting that they are not involved in DAH or are needed in only small numbers (*Figure 11*).

Interestingly, circulating R1 monocytes were numerous in MO-treated B6 mice and pristane-treated B6 μMT and BALB/c mice, but they do not develop DAH (*Figure 11D*). Thus, R1 monocytes were not sufficient to induce DAH. One explanation is that the R1 subset might differ in pristane- vs. MO-treated mice, as suggested by the altered annexin-V staining (*Figure 7D*). Alternatively, the Nr4a1-regulated Ly6C$^{-}$ monocytes might differ. Consistent with previous studies in untreated mice (*Mildner et al., 2017*), Nr4a1-dependent Ly6C$^{-}$CD11b$^{hi}$ (R6) cells in MO-treated mice were CD138$^{-}$ and are likely to be patrolling monocytes monitoring the microvascular endothelium for damage (*Auffray et al., 2007*; *Carlin et al., 2013*). Thus, CD138 expression may be a marker for Ly6C$^{-}$ monocytes that have lost the ability to promote vascular repair. Consistent with this hypothesis, Ly6C$^{-}$CD138$^{+}$ monocytes express higher levels of *Tlr7* and *Treml4* than Ly6C$^{-}$CD138$^{-}$ monocytes and produce large amounts of TNFα in response to R848 (*Figure 7A–C*). Annexin-V staining provided further evidence that circulating monocytes are different in pristane- vs. MO-treated mice (*Figure 7D*). Annexin-V staining was substantially

higher on Ly6C[hi] as well as Ly6C[lo/−] monocytes from pristane- vs. MO-treated mice, suggesting that plasma membrane asymmetry is altered by pristane treatment at an early stage (prior to the differentiation of Ly6C[hi] precursors into Ly6C[−] monocytes), with increased phosphatidylserine (PtdSer) exposure in the outer leaflet. Annexin-V staining of monocytes from pristane-treated mice was uniformly shifted to the right (*Figure 7D*). In contrast, neutrophils from the same mice exhibited two populations of cells, one annexin-V[lo] and the other annexin-V[hi]. The annexin-V[hi] population was more prominent in pristane- vs. MO-treated mice, suggesting that pristane causes neutrophil apoptosis. Staining with an antibody against activated caspase-3 was not increased in monocytes from pristane-treated mice, suggesting that pristane may not induce apoptosis in these cells.

Plasma membrane asymmetry is maintained by flippases, such as ATP11C, and phospholipid scramblases, such as ANO6 (TMEM16F) and XKR8, which help retain PtdSer in the inner leaflet (*Segawa and Nagata, 2015*). In apoptotic cells, cleavage by caspases irreversibly inactivates ATP11C flippase and activates XKR8 scramblase, resulting in increased PtdSer exposure on the cell surface. However, in activated cells, increased intracellular Ca[2+] can reversibly activate TMEM16F and inactivate ATP11C, resulting in transiently increased concentrations of PtdSer in the outer leaflet without apoptosis (*Segawa and Nagata, 2015*; *Doktorova et al., 2020*). Loss of plasma membrane asymmetry plays a role in immune signaling, including the release of cytokine-loaded microvesicles from monocytes (*MacKenzie et al., 2001*). We conclude that pristane may selectively alter the activation state of monocytes, leading to a reversible loss of plasma membrane asymmetry. Increased CD138 expression may be marker for this activated state, which is associated with increased production of inflammatory cytokines (*Figure 7C*), possibly via a mechanism similar to that reported for IL-1β (*MacKenzie et al., 2001*).

## CD138⁺ monocytes are not precursors of peritoneal CD138⁺ Mϕ

Studies in *Ccr2−/−* and *Nr4a1−/−* mice strongly suggested that Ly6C[−]CD138[+] peritoneal Mϕ were not derived from Ly6C[−]CD138[+] monocytes. Migration of Ly6C[hi] monocytes to the peritoneum in pristane-treated mice requires Ccr2 (*Lee et al., 2009*). Consistent with a previous report (*Carlin et al., 2013*), circulating Ly6C[hi] monocytes were reduced by about 50% in pristane-treated *Ccr2−/−* mice but in the peritoneum, both Ly6C[hi] (R1) and Ly6C[−]CD138[+] (R3) Mϕ were nearly absent (*Figure 3B*). The inability of the remaining circulating Ly6C[hi] monocytes to enter the peritoneum suggests that both BM egress and migration to the peritoneum were Ccr2 dependent. Following monocyte/Mϕ depletion with clodronate liposomes, the peritoneum is repopulated by Ly6C[hi] monocytes, which differentiate into Ly6C[−] Mϕ (*Lee et al., 2009*). The absence of both Ly6C[hi] and CD138[+] Mϕ in the peritoneum of *Ccr2−/−* mice suggests that CD138[+] Mϕ are derived from Ly6C[hi] monocyte precursors. Moreover, although Ly6C[−]CD138[+] monocytes were reduced in *Nr4a1−/−* mice, percentages of Ly6C[−]CD138[+] Mϕ were similar (*Figure 4B,C*), confirming that peritoneal CD138[+] Mϕ were not likely to be derived from circulating CD138[+] monocytes.

## What is the role of CD138?

The function of CD138 (syndecan-1, *Sdc1*) in myeloid cells is uncertain. Although the protein is expressed by a subset of Ly6C[−] monocytes and Mϕ, *Sdc1* is not among the genes differentially expressed in Ly6C[hi] vs. Ly6C[−] monocytes from non-pristane-treated-mice (*Mildner et al., 2017*). There are several plausible explanations. First, the antibody we used might cross-react with another target in myeloid cells. However, clone 281-2 is a standard antibody for flow cytometry of mouse CD138 that recognizes the extracellular domain of CD138 on plasma cells, epithelial cells, and early B cells (*Jalkanen et al., 1985*). Although transcriptional profiling of Ly6C[−]CD138[+] monocytes was not done, flow-sorted CD138[+] Mϕ express high levels of *Sdc1* mRNA (*Han et al., 2020*) and others also have reported CD138 mRNA and protein expression in peritoneal Mϕ (*Yeaman and Rapraeger, 1993*). A more likely possibility is that *Sdc1* is not normally expressed in monocytes, but is induced directly or indirectly by pristane.

CD138 is a heparan sulfate-containing membrane proteoglycan that binds extracellular matrix proteins and promotes integrin activation, wound healing, cell adhesion/migration, endocytosis, and fibrosis (*Stepp et al., 2015*). The extracellular domain binds type IV collagen and laminins (*Stepp et al., 2015*; *San Antonio et al., 1994*; *Salmivirta et al., 1994*; *Jung et al., 2009*), which are components of the basement membrane separating the alveolar capillary endothelium and the epithelium

(*Loscertales et al., 2016*). In chronic inflammation, monocytes attach to the vascular endothelium via LFA-1, CX3CR1, and CD11b (*Auffray et al., 2007*; *Gao et al., 2012*), all of which were highly expressed by Ly6C⁻CD138⁺ monocytes (*Figure 5C* and *Figure 6C,D*). They then migrate to the subendothelial space, and reversibly attach to basement membrane laminin via integrins (*Kostidou et al., 2009a*; *Kostidou et al., 2009b*). It will be of interest to investigate whether CD138 expression promotes monocyte migration and/or attachment to the basement membranes of damaged blood vessels.

## Treml4 is transcriptionally regulated by Nr4a1

*Nr4a1* (Nur77) is a member of the nuclear receptor 4A subfamily that regulates monocyte and Mɸ differentiation (*McEvoy et al., 2017*). It is highly expressed in Ly6C⁻ monocytes, which are deficient in *Nr4a1*−/− mice (*Hanna et al., 2011*; *Thomas et al., 2016*). Our data suggest that *Treml4* is regulated by Nr4a1. *Treml4* mRNA and protein levels correlated with Nr4a1 expression in pristane-treated mice and in TLR ligand- or pristane-treated RAW264.7 cells (*Figures 8 and 10*). Conversely, *Nr4a1* knock-down decreased *Treml4* expression in RAW264.7 cells and *Treml4* expression was lower in adherent Mɸ from *Nr4a1*−/− mice vs. controls (*Figure 10*). Although Nr4a1 appears to transcriptionally regulate *Treml4*, additional factors are likely to be involved, since low levels of *Treml4* were expressed in Ly6C⁻CD138⁺ and Ly6C⁻CD138⁻ monocytes and Mɸ from *Nr4a1*−/− mice (*Figure 8*). Relevant to the role of Ly6C⁻ monocytes in monitoring endothelial damage, TremL4 binds late apoptotic and necrotic cells and sensitizes myeloid cells to TLR7 signaling by recruiting MyD88 to endosomes (*Hemmi et al., 2009*; *Ramirez-Ortiz et al., 2015*). Thus, high levels of TremL4 may promote inflammatory cytokine production by monocytes infiltrating the lung in mice with DAH. TremL4 also may play a role in lupus nephritis. *Treml4*−/− MRL/*lpr* mice have lower autoantibody and interferon-α production and improved survival vs. *Treml4*+/+ controls (*Ramirez-Ortiz et al., 2015*), suggesting that Treml4 plays a causal role in lupus nephritis by upregulating TLR7-driven interferon production. Since patrolling monocytes play a role in glomerular inflammation in murine lupus nephritis (*Kuriakose et al., 2019*), Treml4 expression may enhance the inflammatory response to injured glomerular endothelial cells.

Genetic polymorphisms that increase *TREML4* mRNA expression in human peripheral blood cells are associated with the progression and extent of coronary atherosclerotic lesions (*Duarte et al., 2019*). However, a causal relationship has not been established and most evidence suggests that patrolling monocytes are protective in atherosclerosis (*Hanna et al., 2012*; *Narasimhan et al., 2019*). It is possible that upregulation of TLR7 signaling by Treml4 has a dual effect, on one hand promoting vascular integrity by enhancing the removal of damaged endothelial cells, and on the other enhancing vascular inflammation if the endothelial cell damage cannot be resolved.

In summary, DAH, a manifestation of pristane-induced lupus mediated by monocytes, was abolished in the absence of Ly6Chi inflammatory monocytes. However, Ly6Chi monocytes also increased in MO-treated mice without DAH, suggesting that Ly6C⁻ monocytes play a secondary role. Our data suggest that patrolling monocytes from pristane- and MO-treated mice assume alternative, proinflammatory vs. proresolving phenotypes, respectively. Circulating Nr4a1-dependent Ly6C⁻CD138⁻ monocytes in MO-treated mice were associated with resistance to DAH, whereas Ly6C⁻CD138⁺ monocytes were associated with susceptibility. CD138 may be a marker for an activated subtype of patrolling monocytes, as also suggested by the increased annexin-V staining of monocytes from pristane- vs. MO-treated mice. In pristane-treated mice annexin-V staining also was higher in Ly6Chi monocytes, the precursor of Ly6Clo patrolling monocytes. Thus, pristane may alter the precursors of Ly6Clo patrolling monocytes, either in the BM or immediately upon egress of those cells from the marrow. Patrolling Ly6CloCD138⁺annexinVhi monocytes with high Treml4 expression may cause an exaggerated inflammatory response to damaged lung endothelial cells. In contrast, Ly6C⁻CD138⁻annexinVlo monocytes with low Treml4 expression may be less inflammatory, facilitating the repair of lung vascular damage and protecting from DAH.

## Materials and methods

### Mice

C57BL/6 (B6), B6.129S4-*Ccr2*tm1Ifc/J (*Ccr2*−/−), B6.129S2-*Nr4a1*tm1Jmi/J (*Nr4a1*−/−), B6(Cg)-Ifnar1tm1.2Ees/J (*Ifnar1*−/−), B6.129S2-*Ighm*tm1Cgn/J (μMT), and BALB/cJ mice (Jackson Laboratory, Bar Harbor, ME) maintained under specific pathogen-free conditions were injected i.p. with 0.5 ml of pristane

**Table 2.** Murine monoclonal antibodies.

| Specificity (clone) | Fluorochrome* | Source |
|---|---|---|
| CD138 (281-2) | PE; APC; APC-Cy7 | BioLegend |
| CD11b (M1/70) | BV421, Pacific Blue | BioLegend |
| Ly6C (HK1.4) | APC-Cy7; Alexa Fluor 488 | BioLegend |
| Ly6G (1A8) | APC-Cy7; PE | BioLegend |
| CD43 | FITC | BioLegend |
| TREML4 | PE | BioLegend |
| CD115 | APC-Cy7 | BioLegend |
| F4/80 | PE, BV421 | BioLegend |
| CD62L | PE | BD Pharmingen |
| CD64 | PE | BD Pharmingen |
| CCR2 | FITC | BioLegend |
| Tim4 | APC, PE | BioLegend |
| CD11c | PE | BD Pharmingen |
| CX3CR1 (SA011F11) | FITC | BioLegend |
| LFA-1 | PE | BioLegend |
| NR4A1 | PE | Miltenyi |
| TLR7 | PE | BioLegend |
| CD86 | APC-Cy7 | BioLegend |
| TNFα | APC, PE | BioLegend |

*PE, phycoerythrin; APC, allophycocyanin; FITC, fluorescein isothiocyanate; BV, brilliant violet; APC-Cy7, APC/cyanine 7.

(Sigma-Aldrich, St. Louis, MO) or MO (E.R. Squibb & Sons, New Brunswick, NJ) as described (*Zhuang et al., 2017*). There were 5–15 mice female mice age 8–12 weeks per group unless otherwise noted. Experiments were repeated at least twice. Peritoneal exudate cells (PECs) were collected by lavage 14 days later and heparinized blood was collected from the heart 3–14 days after pristane injection. DAH was assessed as described (*Zhuang et al., 2017*). This study followed the recommendations of the Animal Welfare Act and US Government Principles for the Utilization and Care of Vertebrate Animals and was approved by the UF IACUC.

## Flow cytometry

Flow cytometry of peritoneal cells was performed using anti-mouse CD16/32 (Fc Block, BD Biosciences, San Jose, CA) before staining with primary antibody or isotype controls. Peripheral blood (100 μl) was incubated 30 min in the dark with monoclonal antibodies specific for surface markers. After surface staining, cells were fixed/permeabilized for 5 min (Fix-Perm buffer, eBioscience, San Diego, CA), washed and stained intracellularly for 20 min with anti-Nr4a1 monoclonal antibodies. The cells were washed and resuspended in PBS for flow cytometry. Monoclonal antibodies are listed in *Table 2*.

## R848 stimulation of murine monocytes

Blood was collected by cardiac puncture from B6 mice 14 days after pristane treatment. Erythrocytes were lysed with erythrocyte lysis buffer (Qiagen) and washed with PBS. Leukocytes were resuspended in complete Dulbecco's modified Eagle medium (DMEM) + 10% fetal bovine serum in the presence or absence of R848 (1 μg/ml, Sigma-Aldrich) and cultured in a 5% $CO_2$ atmosphere at 37°C for 5 hr. The cells were prepared for flow cytometry as above using anti- Ly6C-FITC, Ly6G-APC-Cy7, CD138-PE,

and CD11b-BV-421 (surface staining) followed by intracellular staining with anti-TNFα-APC antibodies. Ly6G$^+$ cells were gated out and intracellular TNFα staining was measured in CD11b$^+$Ly6C$^{hi}$CD138$^-$, CD11b$^+$Ly6C$^{lo}$CD138$^-$, and CD11b$^+$Ly6C$^{hi}$CD138$^+$ cells (R1, R2, and R3, respectively).

In other experiments, B6 mice were treated with either pristane or MO and 14 days later, blood was collected by cardiac puncture. Erythrocytes were lysed and leukocytes were cultured for 20 hr in low attachment tissue culture plates containing complete DMEM + 10% fetal bovine serum plus R848 (1 μg/ml) or vehicle. Cells were stained with FITC-conjugated anti-Ly6C, APC-conjugated anti-CD138, Bv-421-conjugated anti-CD11b, and APC-Cy7-conjugated anti-Ly6G antibodies followed by staining with PE-conjugated annexin-V (PE Annexin V Apoptosis Detection Kit, BioLegend), using the manufacturer's recommended buffer and protocol.

## RAW264.7 cell culture with TLR ligands

RAW264.7 cells (murine Mϕ cell line, ATCC, Manassas, VA) were seeded in 24-well non-attachment plates (5 × 10$^5$ cells/well) and cultured for 24 hr in complete DMEM + 10% fetal bovine serum with or without LPS (1 μg/ml) or R848 (1 μg/ml). The cells were washed with PBS and analyzed by flow cytometry. Cells were surface stained for TremL4 and CD86 and intracellularly stained for Nr4a1.

For qPCR analysis, RAW264.7 cells were seeded in 6-well plates (3 × 10$^5$ cells/well) and cultured overnight in complete DMEM + 10% FBS. On the following day, the cells were stimulated with LPS (1 μg/ml) or R848 (1 μg/ml) or PBS for 1, 2, 3, 4, 5, or 6 hr. Total RNA was obtained using the QIAamp RNA blood Mini Kit (Qiagen, Germantown, MD), and cDNA was synthesized using the Superscript II First-Strand Synthesis kit (Invitrogen, Carlsbad, CA). SYBR Green qPCR analysis was performed using the CFX Connect Real-Time system (Bio-Rad, Hercules, CA). Gene expression was normalized to 18S RNA and the expression level was calculated using the 2$^{-\Delta\Delta Ct}$ method.

Primer sequences were as follows: *Nr4a1* forward: AGCTTGGGTGTTGATGTTCC; *Nr4a1* reverse: ATGCGATTCTGCAGCTCTT; *Treml4* forward: CTGGAGGTACTCACAACTGCT; *Treml4* reverse: GGCTCTGTCCTACCATTCTATGA; *Tnfa* forward: AGGAGGAGTCTGCGAAGAAGA; *Tnfa* reverse: GGCAGTGGACCATCTAACTCG; 18S rRNA forward: TGCCATCACTGCCATTAAGG; reverse: TGCTTTCCTCAACACCACATG.

## RAW264.7 cell culture with pristane

Pristane or MO (1 ml) was added to PBS (9 ml) containing 100 mg/ml BSA in a 15 ml polypropylene tube and rotated for 48 hr at 4°C. The surface layer of unincorporated hydrocarbon oil was aspirated at the end of the incubation. The amount of pristane incorporated using this method was calculated as described (*Zhuang et al., 2017*) and adjusted to approximately 1 μg/ml. RAW264.7 cells were seeded in 24-well non-attachment plates (3 × 10$^5$ cells/well) and cultured for 24 hr in complete DMEM + 10% FBS with/without hydrocarbon oils at concentrations ranging from 12.5 to 200 ng/ml. Then cells were washed with PBS and analyzed by flow cytometry for surface TremL4 and intracellular Nr4a1.

## Nr4a1 gene silencing

RAW264.7 cells were seeded in 6-well plates (2 × 10$^5$ cells/well) and cultured overnight in antibiotic-free complete DMEM + 10% FBS. On the following day the cells were transfected with Nr4a1 (Nur77) siRNA (sc-36110 from Santa Cruz Biotechnology) or control siRNA-A (sc-37007, Santa Cruz Biotechnology, Dallas RX). siRNA (40 pmol from a 10 μM stock in sterile water) was added to 100 μl of Opti-MEM medium and transfected using siRNA transfection reagent (sc-29528, Santa Cruz Biotechnology) following the manufacturer's instructions. Transfected cells were incubated at 37°C for 48 hr, and then stimulated with LPS (1 μg/ml) for 1, 3, or 6 hr. Total RNA was collected from the cells using the QIAamp RNA blood Mini Kit (Qiagen) and cDNA was synthesized using the Superscript II First-Strand Synthesis kit (Invitrogen). SYBR Green qPCR analysis was performed using the CFX Connect Real-Time system (Bio-Rad). Gene expression was normalized to 18S RNA and the expression level was calculated using the 2$^{-\Delta\Delta Ct}$ method. Primer sequences were as above.

## Peritoneal cell culture with TLR ligands

PECs were harvested from untreated wild-type B6 and *Nr4a1*−/− mice and allowed to adhere to plastic wells (6-well plates, 2 × 10$^6$ cells/well) for 1 hr in AIM V serum free medium (Thermo Fisher Scientific, Waltham, MA). Non-adherent cells were washed off with PBS and the adherent cells were

cultured for 3 hr in complete DMEM + 10% FBS medium containing LPS (1 µg/ml in PBS) or PBS alone. *Nr4a1* and *Treml4* expression was quantified by qPCR as above.

## Statistical analysis

Statistical analyses were performed using Prism 6.0 (GraphPad Software, San Diego, CA). Differences between groups were analyzed by two-sided unpaired Student's *t*-test unless otherwise indicated. Data were expressed as mean ± standard deviation. $p < 0.05$ was considered significant. Experiments were repeated at least twice.

## Acknowledgements

This work was supported by the National Institutes of Health (NIAMS) grant number R01-AR44731 (WR) and by Department of Medicine (HZ).

## Additional information

### Funding

| Funder | Grant reference number | Author |
| --- | --- | --- |
| National Institute of Arthritis and Musculoskeletal and Skin Diseases | R01-AR44731 | Westley H Reeves |

The funders had no role in study design, data collection, and interpretation, or the decision to submit the work for publication.

### Author contributions

Shuhong Han, Conceptualization, Data curation, Formal analysis, Supervision, Validation, Methodology, Project administration, Writing - review and editing; Haoyang Zhuang, Conceptualization, Data curation; Rawad Daniel Arja, Data curation, Software; Westley H Reeves, Formal analysis, Supervision, Funding acquisition, Validation, Investigation, Writing - original draft, Project administration, Writing - review and editing

### Author ORCIDs

Shuhong Han ⓘ http://orcid.org/0000-0002-6951-120X
Rawad Daniel Arja ⓘ http://orcid.org/0000-0003-4555-2098
Westley H Reeves ⓘ http://orcid.org/0000-0002-2924-4549

### Ethics

This study was performed in strict accordance with the recommendations in the Guide for the Care and Use of Laboratory Animals of the National Institutes of Health. All of the animals were handled according to approved Institutional Animal Care and Use Committee (IACUC) protocols (#201704493) of the University of Florida. The protocol was approved by the Committee on the Ethics of Animal Experiments of the University of Florida. Mice were euthanized by exposure to 100% carbon dioxide in an approved administration chamber before removal of tissue. This method is approved by the AVMA panel on Euthanasia. Thoracotomy will be done after $CO_2$ narcosis, and every effort was made to minimize suffering.

### Decision letter and Author response

Decision letter https://doi.org/10.7554/eLife.76205.sa1
Author response https://doi.org/10.7554/eLife.76205.sa2

## Additional files

### Supplementary files

• Transparent reporting form

## Data availability

There are no datasets generated in this current manuscript. The results were put into figures (Figures 1–8).

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
