## [Editor Report]

This is an interesting manuscript describing two subsets of Ly6Clo non-classical monocytes in pristane-treated mice, one CD138- and Nr4a1-independent and the other CD138+ and Nr4a1-dependent. The latter is believed to be produced during pristane-induced microvascular lung injury in an ineffectual effort to maintain vascular integrity in the face of ongoing endothelial damage. This is a model of lung injury with relevance to autoimmunity. Should this CD138+ Nr4a1-dependent population be a hallmark of vascular injury as the authors suggest this work could be relevant for many fields.

---

## [Decision Letter]

**Decision letter after peer review:**

Thank you for submitting your article "Two subsets of circulating Ly6C^lo^ monocytes distinguished by CD138 (syndecan-1) expression and Nr4a1 dependence in pristane-treated mice" for consideration by *eLife*. Your article has been reviewed by 2 peer reviewers, and the evaluation has been overseen by a Reviewing Editor and Paul Noble as the Senior Editor. The reviewers have opted to remain anonymous.

Essential revisions:

The study is believed to be of broad interest if the revisions can be provided for the issues raised by both reviewers.

The role of NR4a1-dependent CD138hi cells in vascular injury, either contributing to persistence of damage or ineffectually attempting to resolve it, remains uncertain, as does the apparently unique role of CD138 on these cells and their relationship to the Ly6Clo, NR4a1-independent CD138lo subset and function of the latter.

However, we do not know what these cells are doing, and how they are related to other Ly6Clo cells. This latter question can be answered with further experimentation -- RNA expression as one example. The former question is more challenging to address, requiring further experiments that might well be beyond scope of the current paper. But it does seem a reasonable question to ask, and one relatively straightforward to answer, how the Ly6Clo populations alike or not, and how they related to Ly6Chi precursors. Such data might well lead to a better understanding of function.

In addition, reviewer 2 has identified some additional issues to be addressed.

*Reviewer #2 (Recommendations for the authors):*

Abstract: The abstract was a bit difficult to get through. It may be best to tell a story with the results that are pertinent, which will flow better.

– Figure 1B-E and Figure 3A-E. This representation of MFI and running stats between different tissues is not ideal, as it doesn't take into account cell preparation, autofluorescence, etc. It would be better to show the data as a fold change in MFI over the MFI of a control cell population deemed negative in each tissue (maybe neutrophils if they show dimmer expression of CD138) to show background MFI.

– Figure 2F. It would be good here to put the neutrophil population on these plots.

– Results: Low TremL4 expression in Nr4a1-/- mice. "Ly6Chi (R1) cells may acquire Treml4 surface staining as they downregulate Ly6C and upregulate CD138 (Figure 6C, lower right)". It is unclear what this statement refers to on the Figure pane.

– Figure 6D. Details for this figure are lacking from the text and the legend. Are these WT mice given pristane? Does this hold up if all samples in 6E are analyzed this way-not just pristane?

– Figure 6D-E,G. Please show representative staining flow plots of NR4A1.

---

## [Author Response]

Essential revisions:The study is believed to be of broad interest if the revisions can be provided for the issues raised by both reviewers.The role of NR4a1-dependent CD138hi cells in vascular injury, either contributing to persistence of damage or ineffectually attempting to resolve it, remains uncertain, as does the apparently unique role of CD138 on these cells and their relationship to the Ly6Clo, NR4a1-independent CD138lo subset and function of the latter.However, we do not know what these cells are doing, and how they are related to other Ly6Clo cells. This latter question can be answered with further experimentation -- RNA expression as one example. The former question is more challenging to address, requiring further experiments that might well be beyond scope of the current paper. But it does seem a reasonable question to ask, and one relatively straightforward to answer, how the Ly6Clo populations alike or not, and how they related to Ly6Chi precursors. Such data might well lead to a better understanding of function.

We carried out extensive new experiments looking at the relationship of the Ly6C^hi^ and Ly6C^lo^ subsets and the role of different subsets of Ly6C^lo^ monocytes in the pathogenesis of lung hemorrhage. The data can be summarized as follows (we also summarize the data in a new Figure 11):

1) In pristane-treated Ccr2-/- mice, circulating Ly6C^hi^ monocytes as well as Ly6C^lo^CD138^+^ monocytes are nearly absent and the mice do not develop hemorrhage (Figure 3, Figure 4E, Figure 11). In contrast, the Ly6C^lo^CD138^-^ monocyte subset is unaffected by the absence of CCR2, and therefore is dispensable for DAH.

2) In pristane-treated Nr4a1-/- mice, Ly6C^hi^ and Ly6C^lo^CD138^-^ monocytes are present in normal numbers, but the number of Ly6C^lo^CD138^+^ is reduced substantially (Figure 4, Fig, 11). Since a) monocyte depletion prevents DAH (see Ref. 17) and b) Nr4a1-/- mice develop hemorrhage whereas Ccr2-/- mice do not (Figure 4E), it is likely that Ly6C^hi^ monocytes are required for DAH. However, the role of Ly6C^lo^CD138^+^ monocytes is less clear because these cells are not completely absent in Nr4a1-/- mice (Figure 4C). Moreover, the number of Ly6C^hi^ monocytes in mineral oil (MO)-treated mice is comparable to that in pristane-treated mice (Figure 5A-B), yet MO-treated mice do not develop DAH. However, the Ly6C^lo^CD138^+^ monocyte subset is nearly absent in MO-treated mice, suggesting that it has a role in DAH.

3) From the data summarized in 1) and 2) above, we conclude that Ly6C^hi^ monocytes are required for DAH to develop, either because they directly cause lung injury or because they are the precursors of Nr4a1-dependent Ly6C^lo^ monocytes (see Refs. 6 and 9), which in pristane-treated mice are CD138^+^. If Ly6C^lo^CD138^+^ monocytes are involved in alveolar hemorrhage, they apparently are not required in large numbers (as suggested by the data in Nr4a1-/- mice).

4) Adding complexity to our data, additional experiments we carried out in response to the reviewers’ comments indicate that both the Nr4a1-dependent Ly6C^lo^ subset and the Ly6C^hi^ subset are different in pristane- vs. MO- treated mice. Single cell RNA-Seq is an obvious approach to further define these differences, but was beyond what could be done in the context of the present study. Instead, we more extensively phenotyped the various monocyte subsets by flow cytometry (see Figures 5-7). In MO-treated B6 mice, pristane-treated BALB/c, and pristane-treated B6 μMT mice, lung hemorrhage does not develop despite the presence of Ly6C^hi^ monocytes. We found a subset of Nr4a1-dependent Ly6C^lo^CD138^-^ monocytes in these DAH-resistant mice, and propose that they may protect the lung from the induction of lung hemorrhage by Ly6C^hi^ monocytes. An alternative explanation is that Ly6C^hi^ monocytes are different in pristane- vs. MO- treated mice. In our previous bulk RNA-Seq studies of Ly6C^hi^ cells from the peritoneum (derived from Ly6C^hi^ monocytes, but undergoing Mϕ differentiation), we did not observe substantial differences in gene expression between MO vs. pristane treated mice. However, in the additional experiments we carried out for the current paper, we found that circulating Ly6C^hi^ as well as Ly6C^lo^ monocytes from pristane-treated mice exhibit higher annexin-V staining than those from MO-treated mice (Figure 7D). We think this shift to the right of annexin-V staining represents increased monocyte activation (rather than early apoptosis) in the pristane-treated group because these cells did not express activated caspase-3. Increased annexin-V staining intensity was seen on all Ly6C^hi^ and Ly6C^lo^ monocytes, but not on the majority of neutrophils, from pristane-treated mice, suggesting that monocyte activation is present immediately upon egress of Ly6C^hi^ monocytes from the bone marrow. It appears to persist after the differentiation of these “inflammatory” Ly6C^hi^ precursors into Ly6C^lo^ monocytes with a patrolling function.

5) The new data may suggest that CD138 on Ly6C^lo^ monocytes (along with increased annexin-V staining) may serve as a marker for cell activation. The activated phenotype of the Nr4a1-dependent Ly6C^lo^CD138^+^ monocyte subset also is supported by the high level of TremL4 expression on these cells (Figure 7B). Consistent with the known enhancement of TLR7 signaling by TremL4, R848-stimulated Ly6C^lo^CD138^+^ monocytes from pristane-treated mice produced as much TNFα as the Ly6C^hi^ monocytes (Figure 7C).

6) The model that has emerged from these studies is fairly complex, and we therefore included a summary figure (Figure 11). Our model proposes that there are two types of Nr4a1-dependent, Ly6C^lo^, patrolling monocytes. In MO-treated B6 mice (as well as pristane-treated μmt and BALB/c mice), they are predominantly CD138^-^ and may promote the repair of lung injury. Conversely, in pristane-treated B6 mice, they are CD138^+^ (and annexin-V^hi^) with an activated (proinflammatory) phenotype that may perpetuate lung injury.

Reviewer #2 (Recommendations for the authors):Abstract: The abstract was a bit difficult to get through. It may be best to tell a story with the results that are pertinent, which will flow better.

We agree, and rewrote the abstract to “tell a story” as recommended by the reviewer. Hopefully it is easier to read now.

– Figure 1B-E and Figure 3A-E. This representation of MFI and running stats between different tissues is not ideal, as it doesn't take into account cell preparation, autofluorescence, etc. It would be better to show the data as a fold change in MFI over the MFI of a control cell population deemed negative in each tissue (maybe neutrophils if they show dimmer expression of CD138) to show background MFI.

Data are expressed as fold change (over neutrophils) as requested. Expressing the data in this manner did not alter the overall conclusions.

– Figure 2F. It would be good here to put the neutrophil population on these plots.

Due to the large number of new figures added in revision (now a total of 11 figures), we elected to omit Figure 2F.

– Results: Low TremL4 expression in Nr4a1-/- mice. "Ly6Chi (R1) cells may acquire Treml4 surface staining as they downregulate Ly6C and upregulate CD138 (Figure 6C, lower right)". It is unclear what this statement refers to on the Figure pane.

We added red arrows in the figure to indicate the Ly6C^hi^ cells (Figure 8C in the revised manuscript) and added additional clarification in the text (p. 14).

– Figure 6D. Details for this figure are lacking from the text and the legend. Are these WT mice given pristane? Does this hold up if all samples in 6E are analyzed this way-not just pristane?

These are pristane-treated mice. This still holds up when the samples from pristane and PBS treated mice (Figure 8E in the revised manuscript) are analyzed in this way (Figure 8B right in the revised manuscript).

– Figure 6D-E,G. Please show representative staining flow plots of NR4A1.

A representative flow staining plot for Nr4a1 is shown in Figure 8D as requested.